# Graph Edit Distance with General Costs Using Neural Set Divergence

**Eeshaan Jain**[*†]    **Indradyumna Roy**[*‡]
**Saswat Meher**[‡]    **Soumen Chakrabarti**[‡]    **Abir De**[‡]
[†]EPFL    [‡]IIT Bombay
eeshaan.jain@epfl.ch
{saswatmeher,soumen,indraroy15,abir}@cse.iitb.ac.in

## Abstract

Graph Edit Distance (GED) measures the (dis-)similarity between two given graphs, in terms of the minimum-cost edit sequence that transforms one graph to the other. However, the exact computation of GED is NP-Hard, which has recently motivated the design of neural methods for GED estimation. However, they do not explicitly account for edit operations with different costs. In response, we propose GRAPHEDX, a neural GED estimator that can work with general costs specified for the four edit operations, *viz.*, edge deletion, edge addition, node deletion and node addition. We first present GED as a quadratic assignment problem (QAP) that incorporates these four costs. Then, we represent each graph as a set of node and edge embeddings and use them to design a family of neural set divergence surrogates. We replace the QAP terms corresponding to each operation with their surrogates. Computing such neural set divergence require aligning nodes and edges of the two graphs. We learn these alignments using a Gumbel-Sinkhorn permutation generator, additionally ensuring that the node and edge alignments are consistent with each other. Moreover, these alignments are cognizant of both the presence and absence of edges between node-pairs. Experiments on several datasets, under a variety of edit cost settings, show that GRAPHEDX consistently outperforms state-of-the-art methods and heuristics in terms of prediction error. The code is available at https://github.com/structlearning/GraphEdX.

## 1 Introduction

The Graph Edit Distance (GED) between a source graph, $G$, and a target graph, $G'$, quantifies the minimum cost required to transform $G$ into a graph isomorphic to $G'$. This transformation involves a sequence of edit operations, which can include node and edge insertions, deletions and substitutions. Each type of edit operation may incur a different and distinctive cost, allowing the GED framework to incorporate domain-specific knowledge. Its flexibility has led to the widespread use of GED for comparing graphs across diverse applications including graph retrieval [5, 6], pattern recognition [47], image and video indexing [51, 49] and chemoinformatics [21]. Because costs for addition and deletion may differ, GED is not necessarily symmetric, *i.e.*, $\text{GED}(G, G') \neq \text{GED}(G', G)$. This flexibility allows GED to model a variety of graph comparison scenarios, such as finding the Maximum Common Subgraph [43] and checking for Subgraph Isomorphism [13]. In general, it is hard to even approximate GED [32]. Recent work [5, 6, 19, 56, 39] has leveraged graph neural networks (GNNs) to build neural models for GED computation, but many of these approaches cannot account for edit operations with different costs. Moreover, several approaches [40, 31, 56, 6] cast GED as the Euclidean distance between graph embeddings, leading to models that are overly attuned to cost-invariant edit sequences.

---

[*]Equal contribution. Eeshaan Jain did this work while at IIT Bombay.

38th Conference on Neural Information Processing Systems (NeurIPS 2024).

## 1.1 Present work

We propose a novel neural model for computing GED, designed to explicitly incorporate the various costs of edit operations. Our contributions are detailed as follows.

**Neural set divergence surrogates for GED** We formulate GED under general (non-uniform) cost as a quadratic assignment problem (QAP) with four asymmetric distance terms representing edge deletion, edge addition, node deletion and node addition. The edge-edit operations involve quadratic dependencies on a node alignment plan — a proposed mapping of nodes from the source graph to the target graph. To avoid the the complexity of QAP [45], we design a family of differentiable set divergence surrogates, which can replace the QAP objective with a more benign one. In this approach, each graph is represented as a set of embeddings of nodes and node-pairs (edges or non-edges). We replace the original QAP distance terms with their corresponding set divergences, and obtain the node alignment from a differentiable alignment generator modeled using a Gumbel-Sinkhorn network. This network produces a soft node permutation matrix based on contextual node embeddings from the graph pairs, enabling the computation of the overall set divergence in a differentiable manner, which facilitates end-to-end training. Our proposed model relies on late interaction, where the interactions between the graph pairs occur only at the final layer, rather than during the embedding computation in the GNN. This supports the indexing of embedding vectors, thereby facilitating efficient retrieval through LSH [25, 24, 12], inverted index [20], graph based ANN [34, 37] *etc.*

**Learning all node-pair representations** The optimal sequence of edits in GED is heavily influenced by the global structure of the graphs. A perturbation in one part of the graph can have cascading effects, necessitating edits in distant areas. To capture this sensitivity to structural changes, we associate both edges as well as non-edges with suitable expressive embeddings that capture the essence of subgraphs surrounding them. Note that the embeddings for non-edges are never explicitly computed during GNN message-passing operations. They are computed only once, after the GNN has completed its usual message-passing through *existing* edges, thereby minimizing additional computational overhead.

**Node-edge consistent alignment** To ensure edge-consistency in the learned node alignment map, we explicitly compute the node-pair alignment map from the node alignment map and then utilize this derived map to compute collective edge deletion and addition costs. More precisely, if $(u, v) \in G$ and $(u', v') \in G'$ are matched, then the nodes $u$ and $v$ are constrained to match with $u'$ and $v'$ (or, $v'$ and $u'$) respectively. We call our neural framework as GRAPHEDX.

Our experiments across several real datasets show that (1) GRAPHEDX outperforms several state-of-the-art methods including those that use early interaction; (2) the performance of current state-of-the-art methods improves significantly when their proposed distance measures are adjusted to reflect GED-specific distances, as in our approach.

## 2 Related work

**Heuristics for Graph Edit Distance** GED was first introduced in [46]. Bunke and Allermann [14] used it as a tool for non exact graph matching. Later on, [13] connected GED with maximum common subgraph estimation. Blumenthal [7] provide an excellent survey. As they suggest, combinatorial heuristics to solve GED predominantly follows three approaches: (1) Linear sum assignment problem with error-correction, which include [27, 41, 53, 55] (2) Linear programming, which predominantly uses standard tools like Gurobi, (3) Local search [42]. However, they can be extremely time consuming, especially for a large number of graph pairs. Among them Zheng et al. [55] operate in our problem setting, where the cost of edits are different across the edit operations, but for the same edit operation, the cost is same across node or node pairs.

**Optimal Transport** In our work, we utilize Graph Neural Networks (GNNs) to represent each graph as a set of node embeddings. This transforms the inherent Quadratic Assignment Problem (QAP) of graph matching into a Linear Sum Assignment Problem (LSAP) on the sets of node embeddings. Essentially, this requires solving an optimal transport problem in the node embedding space. The use of neural surrogates for optimal transport was first proposed by Cuturi [16], who introduced entropy regularization to make the optimal transport objective strictly convex and utilized Sinkhorn iterations [50] to obtain the transport plan. Subsequently, Mena et al. [35] proposed the neural Gumbel Sinkhorn network as a continuous and differentiable surrogate of a permutation matrix, which we incorporate into our model.

In various generative modeling applications, optimal transport costs are used as loss functions, such as in Wasserstein GANs [1, 3]. Computing the optimal transport plan is a significant challenge, with approaches leveraging the primal formulation [52, 33], the dual formulation with entropy regularization [17, 48, 22], or Input Convex Neural Networks (ICNNs) [2].

**Neural graph similarity computation**   Most earlier works on neural graph similarity computation have focused on training with GED values as ground truth [5, 6, 19, 40, 56, 39, 54, 31], while some have used MCS as the similarity measure [6, 5]. Current neural models for GED approximation primarily follow two approaches. The first approach uses a trainable nonlinear function applied to graph embeddings to compute GED [5, 39, 6, 56, 54, 19]. The second approach calculates GED based on the Euclidean distance in the embedding space [31, 40].

Among these models, GOTSIM [19] focuses solely on node insertion and deletion, and computes node alignment using a combinatorial routine that is decoupled from end-to-end training. However, their network struggles with training efficiency due to the operations on discrete values, which are not amenable to backpropagation. With the exception of GREED [40] and Graph Embedding Network (GEN) [31], most methods use early interaction or nonlinear scoring functions, limiting their adaptability to efficient indexing and retrieval pipelines.

## 3   Problem setup

**Notation**   The source graph is denoted by $G = (V, E)$ and the target graph by $G' = (V', E')$. Both graphs are undirected and are padded with isolated nodes to equalize the number of nodes to $N$. The adjacency matrices for $G$ and $G'$ after padding are $\boldsymbol{A}, \boldsymbol{A'} \in \{0, 1\}^{N \times N}$. (Note that we will use $M^\top$, not $M'$, for the transpose of matrix $M$.) The sets of padded nodes in $G$ and $G'$ are denoted by $\mathrm{PaddedNodes}_G$ and $\mathrm{PaddedNodes}_{G'}$ respectively. We construct $\boldsymbol{\eta} \in \{0, 1\}^N$, where $\boldsymbol{\eta}[u] = 0$ if $u \in \mathrm{PaddedNodes}_G$ and 1 otherwise (same for $G'$). The embedding of a node $u \in V$ computed at propagation layer $k$ by the GNN, is represented as $\boldsymbol{x}_k(u)$. Edit operations, denoted by $\mathrm{edit}$, belong to one of four types, *viz.*, (i) node deletion, (ii) node addition, (iii) edge deletion, (iv) edge addition. Each operation $\mathrm{edit}$ is assigned a cost $\mathrm{cost}(\mathrm{edit})$. The node and node-pair alignment maps are described using (hard) permutation matrices $\boldsymbol{P} \in \{0, 1\}^{N \times N}$ and $\boldsymbol{S} \in \{0, 1\}^{\binom{N}{2} \times \binom{N}{2}}$ respectively. Given that the graphs are undirected, node-pair alignment need only be specified across at most $\binom{N}{2}$ pairs. When a hard permutation matrix is relaxed to a doubly-stochastic matrix, we call it a soft permutation matrix. We use $\boldsymbol{P}$ and $\boldsymbol{S}$ to refer to both hard and soft permutations, depending on the context. We denote $\mathbb{P}_N$ as the set of all hard permutation matrices of dimension $N$; $[N]$ as $\{1, \ldots, N\}$ and $\|\boldsymbol{A}\|_{1,1}$ to describe $\sum_{u,v} |\boldsymbol{A}[u, v]|$. For two binary variables $c_1, c_2 \in \{0, 1\}$, we denote $J(c_1, c_2)$ as ($c_1$ XOR $c_2$), *i.e.*, $J(c_1, c_2) = c_1 c_2 + (1 - c_1)(1 - c_2)$.

**Graph edit distance with general cost**   We define an *edit path* as a sequence of edit operations $\boldsymbol{o} = \{\mathrm{edit}_1, \mathrm{edit}_2, \ldots\}$; and $\mathcal{O}(G, G')$ as the set of all possible edit paths that transform the source graph $G$ into a graph isomorphic to the target graph $G'$. Given $\mathcal{O}(G, G')$ and the cost associated with each operation $\mathrm{edit}$, the GED between $G$ and $G'$ is the minimum collective cost across all edit paths in $\mathcal{O}(G, G')$. Formally, we write [14, 7]:

$$\mathrm{GED}(G, G') = \min_{\boldsymbol{o} = \{\mathrm{edit}_1, \mathrm{edit}_2, \ldots\} \in \mathcal{O}(G, G')} \sum_{i \in [|\boldsymbol{o}|]} \mathrm{cost}(\mathrm{edit}_i). \tag{1}$$

In this work, we assume a fixed cost for each of the four types of edit operations. Specifically, we use $a^\ominus$, $a^\oplus$, $b^\ominus$ and $b^\oplus$ to represent the costs for edge deletion, edge addition, node deletion, and node addition, respectively. These costs are not necessarily uniform, in contrast to the assumptions made in previous works [5, 31, 56, 39]. Additional discussion on GED with node substitution in presence of labels can be found in Appendix B.

**Problem statement**   Our objective is to design a neural architecture for predicting GED under a general cost framework, where the edit costs $a^\ominus$, $a^\oplus$, $b^\ominus$ and $b^\oplus$ are not necessarily the same. During the learning stage, these four costs are specified, and remain fixed across all training instances $\mathcal{D} = \{(G_i, G'_i, \mathrm{GED}(G_i, G'_i))\}_{i \in [n]}$. Note that the edit paths are not supervised. Later, given a test instance $G, G'$, assuming the same four costs, the trained system has to predict $\mathrm{GED}(G, G')$.

## 4   Proposed approach

In this section, we first present an alternative formulation of GED as described in Eq. (1), where the edit paths are induced by node alignment maps. Then, we adapt this formulation to develop

GRAPHEDX, a neural set distance surrogate, amenable to end-to-end training. Finally, we present the network architecture of GRAPHEDX.

## 4.1 GED computation using node alignment map

Given the padded graph pair $G$ and $G'$, deleting a node $u \in V$ can be viewed as aligning node $u$ with some padded node $u' \in \text{PaddedNodes}_{G'}$. Similarly, adding a new node $u'$ to $G$ can be seen as aligning some padded node $u \in \text{PaddedNodes}_G$ with node $u' \in V'$. Likewise, adding an edge to $G$ corresponds to aligning a non-edge $(u, v) \notin E$ with an edge $(u', v') \in G'$. Conversely, deleting an edge in $G$ corresponds to aligning an edge $(u, v) \in G$ with a non-edge $(u', v') \notin G'$.

Therefore, $\text{GED}(G, G')$ can be defined in terms of a node alignment map. Let $\Pi_N$ represent the set of all node alignment maps $\pi : [N] \to [N]$ from $V$ to $V'$. Recall that $\boldsymbol{\eta}_G[u] = 0$ if $u \in \text{PaddedNodes}_G$ and 1 otherwise.

$$\min_{\pi \in \Pi_N} \frac{1}{2} \sum_{u,v} \Big( a^{\ominus} \cdot \mathbb{I}\left[(u, v) \in E \wedge (\pi(u), \pi(v)) \notin E'\right] + a^{\oplus} \cdot \mathbb{I}\left[(u, v) \notin E \wedge (\pi(u), \pi(v)) \in E'\right] \Big)$$
$$+ \sum_u \Big( b^{\ominus} \cdot \boldsymbol{\eta}_G[u] \left(1 - \boldsymbol{\eta}_{G'}[\pi(u)]\right) + b^{\oplus} \cdot \left(1 - \boldsymbol{\eta}_G[u]\right) \boldsymbol{\eta}_{G'}[\pi(u)] \Big). \tag{2}$$

In the above expression, the first sum iterates over all pairs of $(u, v) \in [N] \times [N]$ and the second sum iterates over $u \in [N]$. Because both graphs are undirected, the fraction $1/2$ accounts for double counting of the edges. The first and second terms quantify the cost of deleting and adding the edge $(u, v)$ from and to $G$, respectively. The third and the fourth terms evaluate the cost of deleting and adding node $u$ from and to $G$, respectively.

**GED as a Quadratic Assignment Problem**  In its current form, Eq. (2) cannot be immediately adapted to a differentiable surrogate. To circumvent this problem, we provide an equivalent matricized form of Eq. (2), using a hard node permutation matrix $\boldsymbol{P}$ instead of the alignment map $\pi$. We compute the asymmetric distances between $\boldsymbol{A}$ and $\boldsymbol{P}\boldsymbol{A}'\boldsymbol{P}^\top$ and combine them with weights $a^{\ominus}$ and $a^{\oplus}$. Notably, $\text{ReLU}\left(\boldsymbol{A} - \boldsymbol{P}\boldsymbol{A}'\boldsymbol{P}^\top\right)[u, v]$ is non-zero if the edge $(u, v) \in E$ is mapped to a non-edge $(u', v') \in E'$ with $\boldsymbol{P}[u, u'] = \boldsymbol{P}[v, v'] = 1$, indicating deletion of the edge $(u, v)$ from $G$. Similarly, $\text{ReLU}\left(\boldsymbol{P}\boldsymbol{A}'\boldsymbol{P}^\top - \boldsymbol{A}\right)[u, v]$ becomes non-zero if an edge $(u, v)$ is added to $G$. Therefore, for the edit operations involving edges, we have:

$$\mathbb{I}\left[(u, v) \in E \wedge (\pi(u), \pi(v)) \notin E'\right] = \text{ReLU}\left(\boldsymbol{A} - \boldsymbol{P}\boldsymbol{A}'\boldsymbol{P}^\top\right)[u, v], \tag{3}$$

$$\mathbb{I}\left[(u, v) \notin E \wedge (\pi(u), \pi(v)) \in E'\right] = \text{ReLU}\left(\boldsymbol{P}\boldsymbol{A}'\boldsymbol{P}^\top - \boldsymbol{A}\right)[u, v]. \tag{4}$$

Similarly, we note that $\text{ReLU}\left(\boldsymbol{\eta}_G[u] - \boldsymbol{\eta}_{G'}[\pi(u)]\right) > 0$ if $u \notin \text{PaddedNodes}_G$ and $\pi(u) \in \text{PaddedNodes}_{G'}$, which allows us to compute the cost of deleting the node $u$ from $G$. Similarly, we use $\text{ReLU}\left(\boldsymbol{\eta}_{G'}[\pi(u)] - \boldsymbol{\eta}_G[u]\right)$ to account for the addition of the node $u$ to $G$. Formally, we write:

$$\boldsymbol{\eta}_G[u]\left(1 - \boldsymbol{\eta}_{G'}[\pi(u)]\right) = \text{ReLU}\left(\boldsymbol{\eta}_G[u] - \boldsymbol{\eta}_{G'}[\pi(u)]\right), \tag{5}$$

$$\left(1 - \boldsymbol{\eta}_G[u]\right)\boldsymbol{\eta}_{G'}[\pi(u)] = \text{ReLU}\left(\boldsymbol{\eta}_{G'}[\pi(u)] - \boldsymbol{\eta}_G[u]\right). \tag{6}$$

Using Eqs. (3)–(6), we rewrite Eq. (2) as:

$$\text{GED}(G, G') = \min_{\boldsymbol{P} \in \mathbb{P}_N} \frac{a^{\ominus}}{2} \left\|\text{ReLU}\left(\boldsymbol{A} - \boldsymbol{P}\boldsymbol{A}'\boldsymbol{P}^\top\right)\right\|_{1,1} + \frac{a^{\oplus}}{2} \left\|\text{ReLU}\left(\boldsymbol{P}\boldsymbol{A}'\boldsymbol{P}^\top - \boldsymbol{A}\right)\right\|_{1,1}$$
$$+ b^{\ominus} \left\|\text{ReLU}\left(\boldsymbol{\eta}_G - \boldsymbol{P}\boldsymbol{\eta}_{G'}\right)\right\|_1 + b^{\oplus} \left\|\text{ReLU}\left(\boldsymbol{P}\boldsymbol{\eta}_{G'} - \boldsymbol{\eta}_G\right)\right\|_1. \tag{7}$$

The first and the second term denote the collective costs of deletion and addition of edges, respectively. The third and the fourth terms present a matricized representation of Eqs. (5)- (6). The above problem can be viewed as a quadratic assignment problem (QAP) on graphs, given that the hard node permutation matrix $\boldsymbol{P}$ has a quadratic involvement in the first two terms. Note that, in general, $\text{GED}(G, G') \neq \text{GED}(G', G)$. However, the optimal edit paths for these two GED values, encoded by the respective node permutation matrices, are inverses of each other, as formally stated in the following proposition (proven in Appendix B).

**Proposition 1** *Given a fixed set of values of $b^{\ominus}, b^{\oplus}, a^{\ominus}, a^{\oplus}$, let $\boldsymbol{P}$ be an optimal node permutation matrix corresponding to $\text{GED}(G, G')$, computed using Eq. (7). Then, $\boldsymbol{P}' = \boldsymbol{P}^\top$ is an optimal node permutation corresponding to $\text{GED}(G', G)$.*

**Connection to different notions of graph matching**  The above expression of GED can be used to represent various notions of graph matching and similarity measures by modifying the edit costs. These include graph isomorphism, subgraph isomorphism, and maximum common edge subgraph

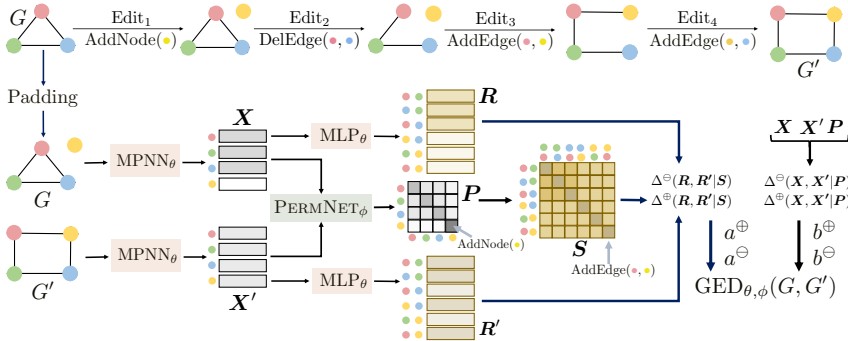

Figure 1: **Top:** Example graphs $G$ and $G'$ are shown with color-coded nodes to indicate alignment corresponding to the optimal edit path transforming $G$ to $G'$. **Bottom:** GRAPHEDX's GED prediction pipeline. $G$ and $G'$ are independently encoded using $\text{MPNN}_\theta$, and then padded with zero vectors to equalize sizes, resulting in contextual node representations $\boldsymbol{X}, \boldsymbol{X}' \in \mathbb{R}^{N \times d}$. For each node-pair, the corresponding embeddings and edge presence information are gathered and fed into $\text{MLP}_\theta$ to obtain $\boldsymbol{R}, \boldsymbol{R}' \in \mathbb{R}^{N(N-1)/2 \times D}$. Simultaneously, $\boldsymbol{X}, \boldsymbol{X}'$ are fed into $\text{PERMNET}_\phi$ to obtain the soft node alignment $\boldsymbol{P}$ (Eq.(16)) which constructs the node-pair alignment matrix $\boldsymbol{S} \in \mathbb{R}^{N(N-1)/2 \times N(N-1)/2}$ as $\boldsymbol{S}[(u,v),(u',v')] = \boldsymbol{P}[u,u']\boldsymbol{P}[v,v'] + \boldsymbol{P}[u,v']\boldsymbol{P}[v,u']$. Finally, $\boldsymbol{X}, \boldsymbol{X}', \boldsymbol{P}$ are used to approximate node insertion and deletion costs, while $\boldsymbol{R}, \boldsymbol{R}', \boldsymbol{S}$ are used to approximate edge insertion and deletion costs. The four costs are summed to give the final prediction $\text{GED}_{\theta,\phi}(G, G')$ (Eq.(8)).

detection. For example, by setting all costs to one, $\text{GED}(G, G') = \min_{\boldsymbol{P}} \frac{1}{2}||\boldsymbol{A} - \boldsymbol{P}\boldsymbol{A}'\boldsymbol{P}^\top||_1 + ||\boldsymbol{\eta}_G - \boldsymbol{P}\boldsymbol{\eta}_{G'}||_1$, which equals zero only when $G$ and $G'$ are isomorphic. Further discussion on this topic is provided in Appendix B.

## 4.2 GRAPHEDX model

Minimizing the objective in Eq. (7) is a challenging problem. In similar problems, recent methods have approximated the hard node permutation matrix $\boldsymbol{P}$ with a soft permutation matrix obtained using Sinkhorn iterations on a neural cost matrix. However, the binary nature of the adjacency matrix and the pad indicator $\boldsymbol{\eta}$ still impede the flow of gradients during training. To tackle this problem, we make relaxations in two key places within each term in Eq. (7), leading to our proposed GRAPHEDX model.

(1) We replace the binary values in $\boldsymbol{\eta}_G, \boldsymbol{\eta}_{G'}, \boldsymbol{A}$ and $\boldsymbol{A}'$ with real values from node and node-pair embeddings: $\boldsymbol{X} \in \mathbb{R}^{N \times d}$ and $\boldsymbol{R} \in \mathbb{R}^{\binom{N}{2} \times D}$. These embeddings are computed using a GNN guided neural module $\text{EMBED}_\theta$ with parameter $\theta$. Since the graphs are undirected, $\boldsymbol{R}$ gathers the embeddings of the unique node-pairs, resulting in $\binom{N}{2}$ rows instead of $N^2$.

(2) We substitute the hard node permutation matrix $\boldsymbol{P}$ with a soft alignment matrix, generated using a differentiable alignment planner $\text{PERMNET}_\phi$ with parameter $\phi$. Here, $\boldsymbol{P}$ is a doubly stochastic matrix, with $\boldsymbol{P}[u, u']$ indicating the "score" or "probability" of aligning $u \mapsto u'$. Additionally, we also compute the corresponding node-pair alignment matrix $\boldsymbol{S}$.

Using these relaxations, we approximate the four edit costs in Eq. (7) with four continuous set distance surrogate functions.

$$\left\|\text{ReLU}\left(\boldsymbol{A} - \boldsymbol{P}\boldsymbol{A}'\boldsymbol{P}^\top\right)\right\|_{1,1} \to \Delta^\ominus(\boldsymbol{R}, \boldsymbol{R}' \,|\, \boldsymbol{S}), \quad \left\|\text{ReLU}\left(\boldsymbol{P}\boldsymbol{A}'\boldsymbol{P}^\top - \boldsymbol{A}\right)\right\|_{1,1} \to \Delta^\oplus(\boldsymbol{R}, \boldsymbol{R}' \,|\, \boldsymbol{S}),$$

$$\left\|\text{ReLU}\left(\boldsymbol{\eta}_G - \boldsymbol{P}\boldsymbol{\eta}_{G'}\right)\right\|_1 \to \Delta^\ominus(\boldsymbol{X}, \boldsymbol{X}' \,|\, \boldsymbol{P}), \quad \left\|\text{ReLU}\left(\boldsymbol{P}\boldsymbol{\eta}_{G'} - \boldsymbol{\eta}_G\right)\right\|_1 \to \Delta^\oplus(\boldsymbol{X}, \boldsymbol{X}' \,|\, \boldsymbol{P}).$$

This gives us an approximated GED parameterized by $\theta$ and $\phi$.

$$\begin{aligned}
\text{GED}_{\theta,\phi}(G, G') = \, &a^\ominus \Delta^\ominus(\boldsymbol{R}, \boldsymbol{R}' \,|\, \boldsymbol{S}) + a^\oplus \Delta^\oplus(\boldsymbol{R}, \boldsymbol{R}' \,|\, \boldsymbol{S}) \\
&+ b^\ominus \Delta^\ominus(\boldsymbol{X}, \boldsymbol{X}' \,|\, \boldsymbol{P}) + b^\oplus \Delta^\oplus(\boldsymbol{X}, \boldsymbol{X}' \,|\, \boldsymbol{P}).
\end{aligned} \tag{8}$$

Note that since $\boldsymbol{R}$ and $\boldsymbol{R}'$ contain the embeddings of each node-pair only once, there is no need to multiply $1/2$ in the first two terms, unlike Eq. (7). Next, we propose three types of neural surrogates to approximate each of the four operations.

**(1) ALIGNDIFF** Given the node-pair embeddings $\boldsymbol{R}$ and $\boldsymbol{R}'$ for the graph pairs $G$ and $G'$, we apply the soft node-pair alignment $\boldsymbol{S}$ to $\boldsymbol{R}'$. We then define the edge edits in terms of asymmetric

differences between $\boldsymbol{R}$ and $\boldsymbol{SR'}$, which serves as a replacement for the corresponding terms in Eq. (7). We write $\Delta^{\ominus}(\boldsymbol{R}, \boldsymbol{R'} \,|\, \boldsymbol{S})$ and $\Delta^{\oplus}(\boldsymbol{R}, \boldsymbol{R'} \,|\, \boldsymbol{S})$ as:

$$\Delta^{\ominus}(\boldsymbol{R}, \boldsymbol{R'} \,|\, \boldsymbol{S}) = \|\mathrm{ReLU}\,(\boldsymbol{R} - \boldsymbol{SR'})\|_{1,1}, \quad \Delta^{\oplus}(\boldsymbol{R}, \boldsymbol{R'} \,|\, \boldsymbol{S}) = \|\mathrm{ReLU}\,(\boldsymbol{SR'} - \boldsymbol{R})\|_{1,1}. \quad (9)$$

Similarly, for the node edits, we can compute $\Delta^{\ominus}(\boldsymbol{X}, \boldsymbol{X'} \,|\, \boldsymbol{P})$ and $\Delta^{\oplus}(\boldsymbol{X}, \boldsymbol{X'} \,|\, \boldsymbol{P})$ as:

$$\Delta^{\ominus}(\boldsymbol{X}, \boldsymbol{X'} \,|\, \boldsymbol{P}) = \|\mathrm{ReLU}\,(\boldsymbol{X} - \boldsymbol{PX'})\|_{1,1}, \quad \Delta^{\oplus}(\boldsymbol{X}, \boldsymbol{X'} \,|\, \boldsymbol{P}) = \|\mathrm{ReLU}\,(\boldsymbol{PX'} - \boldsymbol{X})\|_{1,1}.$$

**(2) DIFFALIGN** In Eq. (9), we first aligned $\boldsymbol{R'}$ using $\boldsymbol{S}$ and then computed the difference from $\boldsymbol{R}$. Instead, here we first computed the pairwise differences between $\boldsymbol{R'}$ and $\boldsymbol{R}$ for all pairs of node-pairs $(e, e')$, and then combine these differences with the corresponding alignment scores $\boldsymbol{S}[e, e']$. We compute the edge edit surrogates $\Delta^{\ominus}(\boldsymbol{R}, \boldsymbol{R'} \,|\, \boldsymbol{S})$ and $\Delta^{\oplus}(\boldsymbol{R}, \boldsymbol{R'} \,|\, \boldsymbol{S})$ as:

$$\Delta^{\ominus}(\boldsymbol{R}, \boldsymbol{R'} \,|\, \boldsymbol{S}) = \sum_{e,e'} \|\mathrm{ReLU}\,(\boldsymbol{R}[e, :] - \boldsymbol{R'}[e', :])\|_1 \, \boldsymbol{S}[e, e'], \quad (10)$$

$$\Delta^{\oplus}(\boldsymbol{R}, \boldsymbol{R'} \,|\, \boldsymbol{S}) = \sum_{e,e'} \|\mathrm{ReLU}\,(\boldsymbol{R'}[e', :] - \boldsymbol{R}[e, :])\|_1 \, \boldsymbol{S}[e, e']. \quad (11)$$

Here, $e$ and $e'$ represent node-pairs, which are not necessarily edges. When the node-pair alignment matrix $\boldsymbol{S}$ is a hard permutation, $\Delta^{\oplus}$ and $\Delta^{\ominus}$ remain the same across ALIGNDIFF and DIFFALIGN (as shown in Appendix B). Similar to Eqs. (10)—(11), we can compute $\Delta^{\ominus}(\boldsymbol{X}, \boldsymbol{X'} \,|\, \boldsymbol{P}) = \sum_{u,u'} \|\mathrm{ReLU}\,(\boldsymbol{X}[u, :] - \boldsymbol{X'}[u', :])\|_1 \, \boldsymbol{P}[u, u']$ and $\Delta^{\oplus}(\boldsymbol{X}, \boldsymbol{X'} \,|\, \boldsymbol{P}) = \sum_{u,u'} \|\mathrm{ReLU}\,(\boldsymbol{X'}[u', :] - \boldsymbol{X}[u, :])\|_1 \, \boldsymbol{P}[u, u']$.

**(3) XOR-DIFFALIGN** As indicated by the combinatorial formulation of GED in Eq. (7), the edit cost of a particular node-pair is non-zero only when an edge is mapped to a non-edge or vice-versa. However, the surrogates for the edge edits in ALIGNDIFF or DIFFALIGN fail to capture this condition because they can assign non-zero costs to the pairs $(e = (u, v), e' = (u', v'))$ even when both $e$ and $e'$ are either edges or non-edges. To address this, we explicitly discard such pairs from the surrogates defined in Eqs. (10)–(11). This is ensured by applying a XOR operator $J(\cdot, \cdot)$ between the corresponding entries in the adjacency matrices, *i.e.*, $\boldsymbol{A}[u, v]$ and $\boldsymbol{A'}[u', v']$, and then multiplying this result with the underlying term. Hence, we write:

$$\Delta^{\ominus}(\boldsymbol{R}, \boldsymbol{R'} \,|\, \boldsymbol{S}) = \sum_{\substack{e=(u,v) \\ e'=(u',v')}} J\big(\boldsymbol{A}[u, v], \boldsymbol{A'}[u', v']\big) \,\|\mathrm{ReLU}\,(\boldsymbol{R}[e, :] - \boldsymbol{R'}[e', :])\|_1 \, \boldsymbol{S}[e, e'], \quad (12)$$

$$\Delta^{\oplus}(\boldsymbol{R}, \boldsymbol{R'} \,|\, \boldsymbol{S}) = \sum_{\substack{e=(u,v) \\ e'=(u',v')}} J\big(\boldsymbol{A}[u, v], \boldsymbol{A'}[u', v']\big) \,\|\mathrm{ReLU}\,(\boldsymbol{R'}[e', :] - \boldsymbol{R}[e, :])\|_1 \, \boldsymbol{S}[e, e']. \quad (13)$$

Similarly, the cost contribution for node operations arises from mapping a padded node to a non-padded node or vice versa. We account for this by multiplying $J(\boldsymbol{\eta}_G[u], \boldsymbol{\eta}_{G'}[u'])$ with each term of $\Delta^{\ominus}(\boldsymbol{X}, \boldsymbol{X'} \,|\, \boldsymbol{P})$ and $\Delta^{\oplus}(\boldsymbol{X}, \boldsymbol{X'} \,|\, \boldsymbol{P})$ computed using DIFFALIGN. Hence, we compute $\Delta^{\ominus}(\boldsymbol{X}, \boldsymbol{X'} \,|\, \boldsymbol{P}) = \sum_{u,u'} J(\boldsymbol{\eta}_G[u], \boldsymbol{\eta}_{G'}[u']) \,\|\mathrm{ReLU}\,(\boldsymbol{X}[u, :] - \boldsymbol{X'}[u', :])\|_1 \, \boldsymbol{P}[u, u']$ and $\Delta^{\oplus}(\boldsymbol{X}, \boldsymbol{X'} \,|\, \boldsymbol{P}) = \sum_{u,u'} J(\boldsymbol{\eta}_G[u], \boldsymbol{\eta}_{G'}[u']) \,\|\mathrm{ReLU}\,(\boldsymbol{X'}[u', :] - \boldsymbol{X}[u, :])\|_1 \, \boldsymbol{P}[u, u']$.

**Comparison between ALIGNDIFF, DIFFALIGN and XOR-DIFFALIGN** ALIGNDIFF and DIFFALIGN become equivalent when $\boldsymbol{S}$ is a hard permutation. However, when $\boldsymbol{S}$ is doubly stochastic, the above three surrogates, ALIGNDIFF, DIFFALIGN and XOR-DIFFALIGN, are not equivalent. As we move from ALIGNDIFF to DIFFALIGN to XOR-DIFFALIGN, we increasingly align the design to the inherent inductive biases of GED, thereby achieving a better representation of its cost structure.

Suppose we are computing the GED between two isomorphic graphs, $G$ and $G'$, with uniform costs for all edit operations. In this scenario, we ideally expect a neural network to consistently output a zero cost. Now consider a proposed soft alignment $\boldsymbol{S}$ which is close to the optimal alignment. Under the ALIGNDIFF design, the aggregated value $\sum_{e'} \boldsymbol{S}[e, e'] \boldsymbol{R'}[e', :]$ — where $e$ and $e'$ represent two edges matched in the optimal alignment — can accumulate over the large number of $N(N-1)/2$ node-pairs. This aggregation leads to high values of $\|\boldsymbol{R}[e, :] - \boldsymbol{SR'}[e', :]\|_1$, implying that ALIGNDIFF captures an aggregate measure of the cost incurred by spurious alignments, but cannot disentangle the effect of individual misalignments, making it difficult for ALIGNDIFF to learn the optimal alignment.

In contrast, the DIFFALIGN approach, which relies on pairwise differences between embeddings to explicitly guide $\boldsymbol{S}$ towards the optimal alignment, significantly ameliorates this issue. For example, in the aforementioned setting of GED with uniform costs, the cost associated with each pairing

$(e, e')$ is explicitly encoded using $||\boldsymbol{R}[e,:] - \boldsymbol{R}'[e',:]||_1$ , and is explicitly set to zero for pairs that are correctly aligned. Moreover, this representation allows DIFFALIGN to isolate the cost incurred by each misalignment, making it easier to train the model to reduce the cost of these spurious matches to zero.

However, DIFFALIGN does not explicitly set edge-to-edge and non-edge-to-non-edge mapping costs to zero, potentially leading to inaccurate GED estimates. XOR-DIFFALIGN addresses these concerns by applying a XOR of the adjacency matrices to the cost matrix, ensuring that non-zero cost is computed only when mapping an edge to a non-edge or vice versa. This resolves the issues in both ALIGNDIFF and DIFFALIGN by focusing on mismatches between edges and non-edges, while disregarding redundant alignments that do not contribute to the GED.

**Amenability to indexing and approximate nearest neighbor (ANN) search.** All of the aforementioned distance surrogates are based on a late interaction paradigm, where the embeddings of $G$ and $G'$ are computed independently of each other before computing the distances $\Delta$. This is particularly useful in the context of graph retrieval, as it allows for the corpus graph embeddings to be indexed a-priori, thereby enabling efficient retrieval of relevant graphs for new queries.

When the edit costs are uniform, our predicted GED (8) becomes symmetric with respect to $G$ and $G'$. In such cases, DIFFALIGN and ALIGNDIFF yield a structure similar to the Wasserstein distance induced by $L_1$ norm. This allows us to leverage ANN techniques like Quadtree or Flowtree [4]. However, while the presence of the XOR operator $J$ within each term in Eq. (12) – (13) of XOR-DIFFALIGN enhances the interaction between $G$ and $G'$, this same feature prevents XOR-DIFFALIGN from being cast to an ANN-amenable setup, unlike DIFFALIGN and ALIGNDIFF.

## 4.3 Network architecture of EMBED$_\theta$ and PERMNET$_\phi$

In this section, we present the network architectures of the two components of GRAPHEDX, *viz.*, EMBED$_\theta$ and PERMNET$_\phi$, as introduced in items (1) and (2) in Section 4.2. Notably, in our proposed graph representation, non-edges and edges alike are embedded as non-zero vectors. In other words, all node-pairs are endowed with non-trivial embeddings. We then explain the design approach for edge-consistent node alignment.

**Neural architecture of EMBED$_\theta$** EMBED$_\theta$ consists of a message passing neural network MPNN$_\theta$ and a decoupled neural module MLP$_\theta$. Given the graphs $G, G'$, MPNN$_\theta$ with $K$ propagation layers is used to iteratively compute the node embeddings $\left\{\boldsymbol{x}_K(u) \in \mathbb{R}^d \,|\, u \in V\right\}$ and $\left\{\boldsymbol{x}'_K(u) \in \mathbb{R}^d \,|\, u \in V'\right\}$, then collect them into $\boldsymbol{X}$ and $\boldsymbol{X}'$ after padding, *i.e.*,

$$\boldsymbol{X} := \{\boldsymbol{x}_K(u) \,|\, u \in [N]\} = \text{MPNN}_\theta(G), \quad \boldsymbol{X}' := \{\boldsymbol{x}'_K(u') \,|\, u' \in [N]\} = \text{MPNN}_\theta(G'). \quad (14)$$

The optimal alignment $\boldsymbol{S}$ is highly sensitive to the global structure of the graph pairs, *i.e.*, $\boldsymbol{S}[e, e']$ can significantly change when we perturb $G$ or $G'$ in regimes distant from $e$ or $e'$. Conventional representations mitigate this sensitivity while training models, by setting non-edges to zero, rendering them invariant to structural changes. To address this limitation, we utilize more expressive graph representations, where non-edges are also embedded using trainable non-zero vectors. This approach allows information to be captured from the structure around the nodes through both edges and non-edges, thereby enhancing the representational capacity of the embedding network. For each node-pair $e = (u, v) \in G$ (and equivalently $(v, u)$), and $e' = (u', v') \in G'$, the embeddings of the corresponding nodes and their connectivity status are concatenated, and then passed through an MLP to obtain the embedding vectors $\boldsymbol{r}(e), \boldsymbol{r}'(e') \in \mathbb{R}^D$. For $e = (u, v) \in G$, we compute $\boldsymbol{r}(e)$ as:

$$\boldsymbol{r}(e) = \text{MLP}_\theta(\boldsymbol{x}_K(u) \,||\, \boldsymbol{x}_K(v) \,||\, \boldsymbol{A}[u,v]) + \text{MLP}_\theta(\boldsymbol{x}_K(v) \,||\, \boldsymbol{x}_K(u) \,||\, \boldsymbol{A}[v,u]). \quad (15)$$

We can compute $\boldsymbol{r}'(e)$ in similar manner. The property $\boldsymbol{r}((u, v)) = \boldsymbol{r}((v, u))$ reflects the undirected property of graph. Finally, the vectors $\boldsymbol{r}(e)$ and $\boldsymbol{r}'(e')$ are stacked into matrices $\boldsymbol{R}$ and $\boldsymbol{R}'$, both with dimensions $\mathbb{R}^{\binom{N}{2} \times D}$. We would like to highlight that $\boldsymbol{r}((u, v))$ or $\boldsymbol{r}'((u', v'))$ are computed only once for all node-pairs, after the MPNN completes its final $K$th layer of execution. The message passing in the MPNN occurs only over edges. Therefore, this approach does not significantly increase the time complexity.

**Neural architecture of PERMNET$_\phi$** The network PERMNET$_\phi$ provides $\boldsymbol{P}$ as a soft node alignment matrix by taking the node embeddings as input, *i.e.*, $\boldsymbol{P} = \text{PERMNET}_\phi(\boldsymbol{X}, \boldsymbol{X}')$. PERMNET$_\phi$ is implemented in two steps. In the first step, we apply a neural network $c_\phi$ on both $\boldsymbol{x}_K$ and $\boldsymbol{x}'_K$, and then compute the normed difference between their outputs to construct the matrix $\boldsymbol{C}$, where $\boldsymbol{C}[u, u'] = \|c_\phi(\boldsymbol{x}_K(u)) - c_\phi(\boldsymbol{x}'_K(u'))\|_1$. Next, we apply iterative Sinkhorn normalizations [16,

[35] on $\exp(-\boldsymbol{C}/\tau)$, to obtain a soft node alignment $\boldsymbol{P}$. Therefore,

$$\boldsymbol{P} = \text{Sinkhorn}\left(\left[\exp\left(-\left\|c_\phi\left(\boldsymbol{x}_K(u)\right) - c_\phi\left(\boldsymbol{x}'_K(u')\right)\right\|_1/\tau\right)\right]_{(u,u')\in[N]\times[N]}\right). \quad (16)$$

Here, $\tau$ is a temperature hyperparameter. In a general cost setting, GED is typically asymmetric, so it may be desirable for $\boldsymbol{C}[u,u']$ to be asymmetric with respect to $\boldsymbol{x}$ and $\boldsymbol{x}'$. However, as noted in Proposition 1, when we compute $\text{GED}(G',G)$, the alignment matrix $\boldsymbol{P}' = \text{PERMNET}_\phi(\boldsymbol{X}',\boldsymbol{X})$ should satisfy the condition that $\boldsymbol{P}' = \boldsymbol{P}^\top$, where $\boldsymbol{P}$ is computed from Eq. (16). The current form of $\boldsymbol{C}$ supports this condition, whereas an asymmetric form might not, as shown in Appendix B.

We construct $\boldsymbol{S} \in \mathbb{R}^{\binom{N}{2}} \times \mathbb{R}^{\binom{N}{2}}$ as follows. Each pair of nodes $(u,v)$ in $G$ and $(u',v')$ in $G'$ can be mapped in two ways, regardless of whether they are edges or non-edges: (1) node $u \mapsto u'$ and $v \mapsto v'$ which is denoted by $\boldsymbol{P}[u,u']\boldsymbol{P}[v,v']$; (2) node $u \mapsto v'$ and $v \mapsto u'$, which is denoted by $\boldsymbol{P}[u,v']\boldsymbol{P}[v,u']$ Combining these two scenarios, we compute the node-pair alignment matrix $\boldsymbol{S}$ as: $\boldsymbol{S}[(u,v),(u',v')] = \boldsymbol{P}[u,u']\boldsymbol{P}[v,v'] + \boldsymbol{P}[u,v']\boldsymbol{P}[v,u']$. This explicit formulation of $\boldsymbol{S}$ from $\boldsymbol{P}$ ensures mutually consistent permutation across nodes and node-pairs.

# 5 Experiments

We conduct extensive experiments on GRAPHEDX to showcase the effectiveness of our method across several real-world datasets, under both uniform and non-uniform cost settings for GED. Additiional experimental results can be found in Appendix D.

## 5.1 Setup

**Datasets** We experiment with seven real-world datasets: Mutagenicity (Mutag) [18], Ogbg-Code2 (Code2) [23], Ogbg-Molhiv (Molhiv) [23], Ogbg-Molpcba (Molpcba) [23], AIDS [36], Linux [5] and Yeast [36]. For each dataset's training, test and validation sets $\mathcal{D}_{\text{split}}$, we generate $\binom{|\mathcal{D}_{\text{split}}|}{2} + |\mathcal{D}_{\text{split}}|$ graph pairs, considering combinations between every two graphs, including self-pairing. We calculate the exact ground truth GED using the F2 solver [29], implemented within GEDLIB [10]. For GED with uniform cost setting, we set the cost values to $b^\ominus = b^\oplus = a^\ominus = a^\oplus = 1$. For GED with non-uniform cost setting, we use $b^\ominus = 3, b^\oplus = 1, a^\ominus = 2, a^\oplus = 1$. Further details on dataset generation and statistics are presented in Appendix C. In the main paper, we present results for the first five datasets under both uniform and non-uniform cost settings for GED. Additional experiments for Linux and Yeast, as well as GED with node label substitutions, are presented in Appendix D.

**Baselines** We compare our approach with nine state-of-the-art methods. These include two variants of GMN [31]: (1) GMN-Match and (2) GMN-Embed; (3) ISONET [44], (4) GREED [40], (5) ERIC [56], (6) SimGNN [5], (7) H2MN [54], (8) GraphSim [6] and (9) EGSC [39]. To compute the GED, GMN-Match, GMN-Embed, and GREED use the Euclidean distance between the vector representation of two graphs. ISONET uses an asymmetric distance specifically tailored to subgraph isomorphism. H2MN is an early interaction network that utilizes higher-order node similarity through hypergraphs. ERIC, SimGNN, and EGSC leverage neural networks to calculate the distance between two graphs. Furthermore, the last three methods predict a score based on the normalized GED in the form of $\exp\left(-2\text{GED}(G,G')/(|V| + |V'|)\right)$. Notably, none of these baseline approaches have been designed to incorporate non-uniform edit costs into their models. To address this limitation, when working with GED under non-uniform cost setting, we include the edit costs as initial features in the graphs for all baseline models. In Appendix D.3, we compare the performance of baselines without cost features.

**Evaluation** Given a dataset $\mathcal{D} = \{(G_i, G'_i, \text{GED}(G_i, G'_i))\}_{i\in[n]}$, we divide it into training, validation and test folds with a split ratio of 60:20:20. We train the models using the Mean Squared Error (MSE) between the predicted GED and the ground truth GED as the loss. For model evaluation, we calculate the Mean Squared Error (MSE) between the actual and predicted GED on the test set. For ERIC, SimGNN and EGSC, we rescale the predicted score to obtain the true (unscaled) GED as $\text{GED}(G,G') = -(|V| + |V'|)\log(s)/2$. In Appendix D, we also report Kendall's Tau (KTau) to evaluate the rank correlation across different experiments.

## 5.2 Results

**Selection of $\Delta^\bullet(\boldsymbol{X}, \boldsymbol{X}' \,|\, \boldsymbol{P})$ and $\Delta^\bullet(\boldsymbol{R}, \boldsymbol{R}' \,|\, \boldsymbol{S})$** We start by comparing the performance of the nine different combinations (three for edge edits, and three for node edits) of our neural distance sur-

rogates from the cartesian space of Edge-{ALIGNDIFF, DIFFALIGN, XOR-DIFFALIGN} × Node-{ALIGNDIFF, DIFFALIGN, XOR-DIFFALIGN}. Table 2 summarizes the results. We make the following observations. (1) The best combinations share the XOR-DIFFALIGN on the edge edit formulation, because, XOR-DIFFALIGN offers more inductive bias, by zeroing the edit cost of aligning an edge to edge and a non-edge to non-edge, as we discussed in Section 4.2. Consequently, one can limit the cartesian space to only three surrogates for node edits, while using XOR-DIFFALIGN as the fixed surrogate for edge edits. (2) There is no clear winner between DIFFALIGN and ALIGNDIFF. GRAPHEDX is chosen from the model which has the lowest validation error, and the numbers in Table 2 are on the test set. Hence, in datasets such as AIDS under uniform cost, or Molhiv under non-uniform cost, the model chosen for GRAPHEDX doesn't have the best test set performance.

| Edge edit | Node edit | GED with uniform cost | | | | | GED with non-uniform cost | | | | |
|---|---|---|---|---|---|---|---|---|---|---|---|
| | | Mutag | Code2 | Molhiv | Molpcba | AIDS | Mutag | Code2 | Molhiv | Molpcba | AIDS |
| DIFFALIGN | DIFFALIGN | 0.579 | 0.740 | 0.820 | 0.778 | 0.603 | 1.205 | 2.451 | 1.855 | 1.825 | 1.417 |
| DIFFALIGN | ALIGNDIFF | 0.557 | 0.742 | 0.806 | 0.779 | 0.597 | 1.211 | 2.116 | 1.887 | 1.811 | 1.319 |
| DIFFALIGN | XOR | 0.538 | 0.719 | 0.794 | 0.777 | 0.580 | 1.146 | 1.896 | 1.802 | 1.822 | 1.381 |
| ALIGNDIFF | DIFFALIGN | 0.537 | 0.513 | 0.815 | 0.773 | 0.606 | 1.185 | 1.689 | 1.874 | 1.758 | 1.391 |
| ALIGNDIFF | ALIGNDIFF | 0.578 | 0.929 | 0.833 | 0.773 | 0.593 | 1.338 | 1.488 | 1.903 | 1.859 | 1.326 |
| ALIGNDIFF | XOR | 0.533 | 0.826 | 0.812 | 0.780 | 0.575 | 1.196 | 1.741 | 1.870 | 1.815 | 1.374 |
| XOR | ALIGNDIFF | 0.492 | 0.429 | 0.788 | 0.766 | 0.565 | 1.134 | 1.478 | 1.872 | 1.742 | 1.252 |
| XOR | DIFFALIGN | 0.510 | 0.634 | 0.781 | 0.765 | 0.574 | 1.148 | 1.489 | 1.804 | 1.757 | 1.340 |
| XOR | XOR | 0.530 | 1.588 | 0.807 | 0.764 | 0.564 | 1.195 | 2.507 | 1.855 | 1.677 | 1.319 |
| GRAPHEDX | | 0.492 | 0.429 | 0.781 | 0.764 | 0.565 | 1.134 | 1.478 | 1.804 | 1.677 | 1.252 |

Table 2: Prediction error measured in terms of MSE of the nine combinations of our neural set distance surrogate across five datasets on test set, for GED with uniform costs and non-uniform costs. For GED with uniform (non-uniform) costs we have $b^{\ominus} = b^{\oplus} = a^{\ominus} = a^{\oplus} = 1$ ($b^{\ominus} = 3, b^{\oplus} = 1, a^{\ominus} = 2, a^{\oplus} = 1$.) The GRAPHEDX model was selected based on the lowest MSE on the validation set, and we report the results of the MSE on the test set. Green ( yellow) numbers report the best (second best) performers.

**Comparison with baselines** We compare the performance of GRAPHEDX against all state-of-the-art baselines for GED with both uniform and non-uniform costs. Table 3 summarizes the results. We make the following observations. (1) GRAPHEDX outperforms all the baselines by a significant margin. For GED with uniform costs, this margin often goes as high as 15%. This advantage becomes even more pronounced for GED with non-uniform costs, where our method outperforms the baselines by a margin as high as 30%, as seen in Code2. (2) There is no clear second-best method. Among the baselines, EGSC and ERIC each outperforms the others in two out of five datasets for both uniform and non-uniform cost settings. Also, EGSC demonstrates competitive performance in AIDS.

| | GED with uniform cost | | | | | GED with non-uniform cost | | | | |
|---|---|---|---|---|---|---|---|---|---|---|
| | Mutag | Code2 | Molhiv | Molpcba | AIDS | Mutag | Code2 | Molhiv | Molpcba | AIDS |
| GMN-Match [31] | 0.797 | 1.677 | 1.318 | 1.073 | 0.821 | 69.210 | 13.472 | 76.923 | 23.985 | 31.522 |
| GMN-Embed [31] | 1.032 | 1.358 | 1.859 | 1.951 | 1.044 | 72.495 | 13.425 | 78.254 | 28.437 | 33.221 |
| ISONET [44] | 1.187 | 0.879 | 1.354 | 1.106 | 1.640 | 3.369 | 3.025 | 3.451 | 2.781 | 5.513 |
| GREED [40] | 1.398 | 1.869 | 1.708 | 1.550 | 1.004 | 68.732 | 11.095 | 78.300 | 26.057 | 34.354 |
| ERIC [56] | 0.719 | 1.363 | 1.165 | 0.862 | 0.731 | 1.981 | 12.767 | 3.377 | 2.057 | 1.581 |
| SimGNN [5] | 1.471 | 2.667 | 1.609 | 1.456 | 1.455 | 4.747 | 5.212 | 4.145 | 3.465 | 4.316 |
| H2MN [54] | 1.278 | 7.240 | 1.521 | 1.402 | 1.114 | 3.413 | 9.435 | 3.782 | 3.396 | 3.105 |
| GraphSim [6] | 2.005 | 3.139 | 2.577 | 1.656 | 1.936 | 5.370 | 7.405 | 6.643 | 3.928 | 5.266 |
| EGSC [39] | 0.765 | 4.165 | 1.138 | 0.938 | 0.627 | 1.758 | 3.957 | 2.371 | 2.133 | 1.693 |
| GRAPHEDX | 0.492 | 0.429 | 0.781 | 0.764 | 0.565 | 1.134 | 1.478 | 1.804 | 1.677 | 1.252 |

Table 3: Prediction error measured in terms of MSE of GRAPHEDX and all the state-of-the-art baselines across five datasets on test set, for GED with uniform costs and non-uniform costs. For GED with uniform (non-unfiform) costs we have $b^{\ominus} = b^{\oplus} = a^{\ominus} = a^{\oplus} = 1$ ($b^{\ominus} = 3, b^{\oplus} = 1, a^{\ominus} = 2, a^{\oplus} = 1$.) GRAPHEDX represents the best model based on the validation set from the cartesian space of Edge-{ALIGNDIFF, DIFFALIGN, XOR-DIFFALIGN} × Node-{ALIGNDIFF, DIFFALIGN, XOR-DIFFALIGN}. Green ( yellow) numbers report the best (second best) performers.

**Impact of cost-guided GED** Among the baselines, GMN-Match, GMN-Embed and GREED compute GED using the euclidean distance between the graph embeddings, *i.e.*, $\text{GED}(G, G') = \|\boldsymbol{x}_G - \boldsymbol{x}_{G'}\|_2$, whereas we compute it by summing the set distance surrogates between the node and edge embedding sets. To understand the impact of our cost guided distance, we adapt it to the graph-level embeddings used by the above three baselines as follows: $\text{GED}(G, G') = \frac{b^{\ominus} + a^{\ominus}}{2} \|\text{ReLU}(\boldsymbol{x}_G - \boldsymbol{x}_{G'})\|_1 + \frac{b^{\oplus} + a^{\oplus}}{2} \|\text{ReLU}(\boldsymbol{x}_{G'} - \boldsymbol{x}_G)\|_1$. Table 4 summarizes the results in

| | Uniform cost | | | Non-uniform cost | | |
|---|---|---|---|---|---|---|
| | Mutag | Code2 | Molhiv | Mutag | Code2 | Molhiv |
| GMN-Match | 0.797 | 1.677 | 1.318 | 69.210 | 13.472 | 76.923 |
| GMN-Match * | **0.654** | **0.960** | **1.008** | **1.592** | **2.906** | **2.162** |
| GMN-Embed | 1.032 | 1.358 | 1.859 | 72.495 | 13.425 | 78.254 |
| GMN-Embed * | **1.011** | **1.179** | **1.409** | **2.368** | **3.272** | **3.413** |
| GREED | **1.398** | 1.869 | 1.708 | 68.732 | 11.095 | 78.300 |
| GREED * | 2.133 | **1.850** | **1.644** | **2.456** | **5.429** | **3.827** |
| GRAPHEDX | 0.492 | 0.429 | 0.781 | 1.134 | 1.478 | 1.804 |

Table 4: Impact of cost guided distance in terms of MSE; * represents the variant of the baseline with cost-guided distance. Green (**bold**) shows the best among all methods (only baselines).

| | Mutag | Code2 | Molhiv | Molpcba | AIDS |
|---|---|---|---|---|---|
| GMN-Match | 1.057 | 5.224 | 1.388 | 1.432 | 0.868 |
| GMN-Embed | 2.159 | 4.070 | 3.523 | 4.657 | 1.818 |
| ISONET | 0.876 | 1.129 | 1.617 | 1.332 | 1.142 |
| GREED | 2.876 | 4.983 | 2.923 | 3.902 | 2.175 |
| ERIC | 0.886 | 6.323 | 1.537 | 1.278 | 1.602 |
| SimGNN | 1.160 | 5.909 | 1.888 | 2.172 | 1.418 |
| H2MN | 1.277 | 6.783 | 1.891 | 1.666 | 1.290 |
| GraphSim | 1.043 | 4.708 | 1.817 | 1.748 | 1.561 |
| EGSC | 0.776 | 8.742 | 1.273 | 1.426 | 1.270 |
| GRAPHEDX | 0.441 | 0.820 | 0.792 | 0.846 | 0.538 |

Table 5: MSE for different methods with unit node substitution cost in uniform cost setting. Green (yellow) show (second) best method.

terms of MSE, which shows that (1) our set-divergence-based cost guided distance reduces the MSE by a significant margin in most cases (2) the margin of improvement is more prominent with GED involving non-uniform costs, where the modeling of specific cost values is crucial (3) GRAPHEDX outperforms the baselines even after changing their default distance to our cost guided distance.

**Performance for GED under node substitution cost** The scoring function in Eq. 8 can also be extended to incorporate node label substitution cost, which has been described in Appendix B. Here, we compare the performance of our model with the baselines in terms of MSE where we include node substitution cost $b^{\sim}$, with cost setting as $b^{\ominus} = b^{\oplus} = b^{\sim} = a^{\ominus} = a^{\oplus} = 1$. In Table 5, we report the results across 5 datasets equipped with node labels, passed as one-hot encoded node features. We observe that (1) our model outperforms all other baselines across all datasets by significant margin; (2) there is no clear second winner but ERIC, EGSC and ISONET performs better than the others.

**Benefits of using all node-pairs representation** In Table 6, we compare against (i) Edge-only (edge → edge): where we only consider the edges that are present, resulting in $S$ being an edge-alignment matrix, and $R, R' \in \mathbb{R}^{\max(|E|,|E'|) \times D}$ (ii) Edge-only (pair → pair): In this variant, the embeddings of the non-edges in $R, R' \in \mathbb{R}^{N(N-1)/2 \times D}$ are explicitly set to zero. in terms of MSE, which show that (1) both these sparse representations perform significantly worse compared to our method using non-trivial representations for both edges and non-edges, and (2) Edge-only (edge → edge)

| | Mutag | Code2 | Molhiv |
|---|---|---|---|
| Edge-only (edge → edge) | 0.566 | 0.683 | 0.858 |
| Edge-only (pair → pair) | 0.596 | 0.760 | 0.862 |
| GRAPHEDX | 0.492 | 0.429 | 0.781 |

Table 6: Comparison of variants of edge representation under uniform cost setting. Green (yellow) numbers report the best (second best) performers.

performs better than Edge-only (pair → pair). This underscores the importance of explicitly modeling trainable non-edge embeddings to capture the sensitivity of GED to global graph structure.

## 6 Conclusion

Our work introduces a novel neural model for computing GED that explicitly incorporates general costs of edit operations. By leveraging graph representations that recognize both edges and non-edges, together with the design of suitable set distance surrogates, we achieve a more robust neural surrogate for GED. Our experiments demonstrate that this approach outperforms state-of-the-art methods, especially in settings with general edit costs, providing a flexible and effective solution for a range of applications. Future work could focus on extending the GED formulation to richly-attributed graphs by modeling the structure of edit operations and the similarity of all node-pair features.

**Limitations** Our neural model for GED showcases significant improvements in accuracy and flexibility for modeling edit costs. However, there are some limitations to consider. (1) While computing graph representations over $\binom{N}{2} \times \binom{N}{2}$ node-pairs does not require additional parameters due to parameter-sharing, it does demand significant memory resources. This could pose challenges, especially with larger-sized graphs. (2) The assumption of fixed edit costs across all graph pairs within a dataset might not reflect real-world scenarios where costs vary based on domain-specific factors and subjective human relevance judgements. This calls for more specialized approaches to accurately model the impact of each edit operation, which may differ across node pairs.

**Acknowledgements** Indradyumna acknowledges Qualcomm Innovation Fellowship, Abir and Soumen acknowledge grants from Amazon, Google, IBM and SERB.

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

# Graph Edit Distance with General Costs
# Using Neural Set Divergence
# (Appendix)

## Contents

# A   Broader impact

Graphs serve as powerful representations across diverse domains, capturing complex relationships and structural notions inherent in various systems. From biological networks to social networks, transportation networks, and supply chains, graphs provide a versatile framework for modeling interactions between interconnected entities. In domains where structure-similarity based applications are prevalent, GED emerges as a valuable and versatile tool.

For example, in bio-informatics, molecular structures can naturally be represented as graphs. GED computation expedites tasks such as drug discovery, protein-protein interaction modeling, and molecular similarity analysis by identifying structurally similar molecular compounds. Similarly, in social network analysis, GED can measure similarities between user interactions, aiding in friend recommendation systems or community detection tasks. In transportation networks, GED-based tools assess similarity between road networks for route planning or traffic optimizations. Further applications include learning to edit scene graphs, analyzing gene regulatory pathways, fraud detection, and more

Moreover, our proposed variations of GED, particularly those amenable to hashing, find utility in retrieval based setups. In various information retrieval systems, hashed graph representations can be used to efficiently index and retrieve relevant items using our GED based scores. Such applications include image retrieval from image databases where images are represented as scene graphs, retrieval of relevant molecules from molecular databases, *etc*.

Furthermore, our ability to effectively model different edit costs in GED opens up new possibilities in various applications. In recommendation systems, it can model user preferences of varying importance, tailoring recommendations based on user-specific requirements or constraints. Similarly, in image or video processing, different types of distortions may have varying impacts on perceptual quality, and GED with adaptive costs can better assess similarity. In NLP tasks such as text similarity understanding and document clustering, assigning variable costs to textual edits corresponding to word insertion, deletions or substitutions, provides a more powerful framework for measuring textual similarity, improving performance in downstream tasks such as plagiarism detection, summarization, *etc*.

Lastly, and most importantly, the design of our model encourages interpretable alignment-driven justifications, thereby promoting transparency and reliability while minimizing potential risks and negative impacts, in high stake applications like drug discovery.

# B  Discussion on our proposed formulation of GED

## B.1  Modification of scoring function from label substitution

To incorporate the effect of node substitution into account when formulating the GED, we first observe that the effect of node substitution cost $b^\sim$ only comes into account when a non-padded node maps to a non-padded node. In all other cases, when a node is deleted or inserted, we do not additionally incur any substitution costs. Note that, we consider the case when node substitution cannot be replaced by node addition and deletion, *i.e.*, $b^\sim \leq b^\ominus + b^\oplus$. Such a constraint on costs has uses in multiple applications [9, 38]. Let $\mathcal{L}$ denote the set of node labels, and $\ell(u), \ell'(u') \in \mathcal{L}$ denote the node label corresponding to nodes $u$ and $u'$ in $G$ and $G'$ respectively. We construct the node label matrix $\boldsymbol{L}$ for $G$ as follows: $\boldsymbol{L} \in \{0,1\}^{N \times |\mathcal{L}|}$, such that $\boldsymbol{L}[i,:] = \texttt{one\_hot}(\ell(i))$, *i.e.*, $\boldsymbol{L}$ is the one-hot indicator matrix for the node labels, which each row corresponding to the one-hot vector of the label. Similarly, we can construct $\boldsymbol{L}'$ for $G'$. Then, the distance between labels of two nodes $u \in V$ and $u' \in V'$ can be given as $\|\boldsymbol{L}[u,:] - \boldsymbol{L}'[u',:]\|_1$. To ensure that only valid node to node mappings contribute to the cost, we multiply the above with $\Lambda(u,u') = \text{AND}(\boldsymbol{\eta}_G[u], \boldsymbol{\eta}_{G'}[u'])$. This allows us to write the expression for GED with node label substitution cost as

$$\text{GED}(G,G') = \min_{\boldsymbol{P} \in \mathbb{P}_N} \frac{a^\ominus}{2} \left\|\text{ReLU}\left(\boldsymbol{A} - \boldsymbol{P}\boldsymbol{A}'\boldsymbol{P}^\top\right)\right\|_{1,1} + \frac{a^\oplus}{2}\left\|\text{ReLU}\left(\boldsymbol{P}\boldsymbol{A}'\boldsymbol{P}^\top - \boldsymbol{A}\right)\right\|_{1,1}$$
$$+ b^\ominus \left\|\text{ReLU}\left(\boldsymbol{\eta}_G - \boldsymbol{P}\boldsymbol{\eta}_{G'}\right)\right\|_1 + b^\oplus \left\|\text{ReLU}\left(\boldsymbol{P}\boldsymbol{\eta}_{G'} - \boldsymbol{\eta}_G\right)\right\|_1$$
$$+ b^\sim \underbrace{\sum_{u,u'} \Lambda(u,u')\left\|\boldsymbol{L}[u,:] - \boldsymbol{L}[u',:]\right\|_1 \boldsymbol{P}[u,u']}_{\Delta^\sim(\boldsymbol{L}, \boldsymbol{L}'|\boldsymbol{P})}$$

We can design a neural surrogate for above in the same way as done in Section 4.2, and write

$$\text{GED}_{\theta,\phi}(G,G') = a^\ominus \Delta^\ominus(\boldsymbol{R}, \boldsymbol{R}' \mid \boldsymbol{S}) + a^\oplus \Delta^\oplus(\boldsymbol{R}, \boldsymbol{R}' \mid \boldsymbol{S})$$
$$+ b^\ominus \Delta^\ominus(\boldsymbol{X}, \boldsymbol{X}' \mid \boldsymbol{P}) + b^\oplus \Delta^\oplus(\boldsymbol{X}, \boldsymbol{X}' \mid \boldsymbol{P})$$
$$+ b^\sim \Delta^\sim(\boldsymbol{L}, \boldsymbol{L}'|\boldsymbol{P}) \tag{17}$$

In this case, to account for node substitutions in the proposed permutation, we use $\boldsymbol{L}[u,:]$ and $\boldsymbol{L}'[u',:]$ as the features for node $u$ in $G$ and node $u'$ in $G'$, respectively. We present the comparison of our method including subsitution cost with state-of-the-art baselines in Appendix D.

## B.2  Proof of Proposition 1

**Proposition**  Given a fixed set of values of $b^\ominus, b^\oplus, a^\ominus, a^\oplus$, let $\boldsymbol{P}$ be an optimal node permutation matrix corresponding to $\text{GED}(G,G')$, computed using Eq. (7). Then, $\boldsymbol{P}' = \boldsymbol{P}^\top$ is an optimal node permutation corresponding to $\text{GED}(G',G)$.

*Proof:*  Noticing that $\text{ReLU}(c-d) = \max(c,d) - d$, we can write

$$\left\|\text{ReLU}\left(\boldsymbol{A} - \boldsymbol{P}\boldsymbol{A}'\boldsymbol{P}^\top\right)\right\|_{1,1} = \left\|\max(\boldsymbol{A}, \boldsymbol{P}\boldsymbol{A}'\boldsymbol{P}^\top) - \boldsymbol{P}\boldsymbol{A}'\boldsymbol{P}^\top\right\|_{1,1}$$
$$= \left\|\max(\boldsymbol{A}, \boldsymbol{P}\boldsymbol{A}'\boldsymbol{P}^\top)\right\|_{1,1} - 2|E'|$$

The last equality follows since $\max(\boldsymbol{A}, \boldsymbol{P}\boldsymbol{A}'\boldsymbol{P}^\top) \geq \boldsymbol{P}\boldsymbol{A}'\boldsymbol{P}^\top$ element-wise, and $\left\|\boldsymbol{P}\boldsymbol{A}'\boldsymbol{P}^\top\right\|_{1,1} = \|\boldsymbol{A}'\|_{1,1} = 2|E'|$. Similarly, we can rewrite $\left\|\text{ReLU}\left(\boldsymbol{P}\boldsymbol{A}'\boldsymbol{P}^\top - \boldsymbol{A}\right)\right\|_{1,1}, \|\text{ReLU}\left(\boldsymbol{\eta}_G - \boldsymbol{P}\boldsymbol{\eta}_{G'}\right)\|_1$, and $\|\text{ReLU}\left(\boldsymbol{P}\boldsymbol{\eta}_{G'} - \boldsymbol{\eta}_G\right)\|_1$, and finally rewrite Eq. (7) as

$$\text{GED}(G,G') = \min_{\boldsymbol{P} \in \mathbb{P}_N} \frac{a^\oplus + a^\ominus}{2} \left\|\max(\boldsymbol{A}, \boldsymbol{P}\boldsymbol{A}'\boldsymbol{P}^\top)\right\|_{1,1} - a^\ominus|E'| - a^\oplus|E|$$
$$+ \frac{b^\oplus + b^\ominus}{2}\left\|\max(\boldsymbol{\eta}_G, \boldsymbol{P}\boldsymbol{\eta}_{G'})\right\|_1 - b^\ominus|V'| - b^\oplus|V| \tag{18}$$

$$\text{GED}(G',G) = \min_{\boldsymbol{P} \in \mathbb{P}_N} \frac{a^\oplus + a^\ominus}{2} \left\|\max(\boldsymbol{A}', \boldsymbol{P}\boldsymbol{A}\boldsymbol{P}^\top)\right\|_{1,1} - a^\ominus|E| - a^\oplus|E'|$$
$$+ \frac{b^\oplus + b^\ominus}{2}\left\|\max(\boldsymbol{\eta}_{G'}, \boldsymbol{P}\boldsymbol{\eta}_G)\right\|_1 - b^\ominus|V| - b^\oplus|V'| \tag{19}$$

We can rewrite the max term as follows:

$$\left\|\max(\boldsymbol{A}, \boldsymbol{P}\boldsymbol{A}'\boldsymbol{P}^\top)\right\|_{1,1} = \sum_{u,v} \max(\boldsymbol{A}, \boldsymbol{P}\boldsymbol{A}'\boldsymbol{P}^\top)[u,v]$$

$$= \sum_{u,v} \max(\boldsymbol{P}\boldsymbol{P}^\top \boldsymbol{A}\boldsymbol{P}\boldsymbol{P}^\top, \boldsymbol{P}\boldsymbol{A}'\boldsymbol{P}^\top)[u,v]$$

$$= \sum_{u,v} \boldsymbol{P}\max(\boldsymbol{P}^\top \boldsymbol{A}\boldsymbol{P}, \boldsymbol{A}')\boldsymbol{P}^\top[u,v]$$

$$= \sum_{u,v} \max(\boldsymbol{P}^\top \boldsymbol{A}\boldsymbol{P}, \boldsymbol{A}')[u,v]$$

$$= \left\|\max(\boldsymbol{P}^\top \boldsymbol{A}\boldsymbol{P}, \boldsymbol{A}')\right\|_{1,1} = \left\|\max(\boldsymbol{A}', \boldsymbol{P}^\top \boldsymbol{A}\boldsymbol{P})\right\|_{1,1}$$

Similarly we can re write $\left\|\max(\boldsymbol{\eta}_G, \boldsymbol{P}\boldsymbol{\eta}_{G'})\right\|_1$ as $\left\|\max(\boldsymbol{\eta}_{G'}, \boldsymbol{P}^\top \boldsymbol{\eta}_G)\right\|_1$. Given a fixed set of cost function $b^\ominus, b^\oplus, a^\ominus, a^\oplus$, the terms containing $|E'|, |E|, |V'|, |V|$ are constant and do not affect choosing an optimal $\boldsymbol{P}$. Let $C = -a^\ominus|E'| - a^\oplus|E| - b^\ominus|V| - b^\oplus|V'|$, Using the above equations, we can write:

$$\frac{a^\oplus + a^\ominus}{2} \left\|\max(\boldsymbol{A}, \boldsymbol{P}\boldsymbol{A}'\boldsymbol{P}^\top)\right\|_{1,1} + \frac{b^\oplus + b^\ominus}{2} \left\|\max(\boldsymbol{\eta}_G, \boldsymbol{P}\boldsymbol{\eta}_{G'})\right\|_1$$

$$= \frac{a^\oplus + a^\ominus}{2} \left\|\max(\boldsymbol{A}', \boldsymbol{P}^\top \boldsymbol{A}\boldsymbol{P})\right\|_{1,1} + \frac{b^\oplus + b^\ominus}{2} \left\|\max(\boldsymbol{\eta}_{G'}, \boldsymbol{P}^\top \boldsymbol{\eta}_G)\right\|_1$$

Let the first term be $\rho(G, G' \,|\, \boldsymbol{P})$. Then second term can be expressed as $\rho(G', G \,|\, \boldsymbol{P}^\top)$ and $\rho(G, G' \,|\, \boldsymbol{P}) = \rho(G', G \,|\, \boldsymbol{P}^\top)$ for all $\boldsymbol{P} \in \mathbb{P}_N$. If $\boldsymbol{P}$ is the optimal solution of $\min_{\boldsymbol{P} \in \mathbb{P}_N} \rho(G, G' \,|\, \boldsymbol{P})$ then, $\rho(G', G \,|\, \boldsymbol{P}^\top) = \rho(G, G' \,|\, \boldsymbol{P}) \leq \rho(G, G' \,|\, \widetilde{\boldsymbol{P}}^\top) = \rho(G', G \,|\, \widetilde{\boldsymbol{P}})$ for any permutation $\widetilde{\boldsymbol{P}}$. Hence, $\boldsymbol{P}' = \boldsymbol{P}^\top \in \mathbb{P}_N$ is one optimal permutation for $\text{GED}(G', G)$.

### B.3 Connections with other notions of graph matching

*Graph isomorphism:* When we set all costs to zero, we can write that $\text{GED}(G, G') = \min_{\boldsymbol{P}} 0.5 \left\|\boldsymbol{A} - \boldsymbol{P}\boldsymbol{A}'\boldsymbol{P}^\top\right\|_{1,1} + \left\|\boldsymbol{\eta}_G - \boldsymbol{P}\boldsymbol{\eta}_{G'}\right\|_1$. In such a scenario, $\text{GED}(G, G')$ is symmetric, *i.e.*, $\text{GED}(G', G) = \text{GED}(G, G')$ and it becomes zero only when $G$ and $G'$ are isomorphic.

*Subgraph isomorphism:* Assume $b^\ominus = b^\oplus = 0$. Then, if we set the cost of edge addition to be arbitrarily small as compared to the cost of edge deletion, *i.e.*, $a^\oplus \ll a^\ominus$. This yields $\text{GED}(G, G') = \min_{\boldsymbol{P}}(b^\ominus \sum_{u,v} \text{ReLU}\left(\boldsymbol{A} - \boldsymbol{P}\boldsymbol{A}'\boldsymbol{P}^\top\right)[u,v])$, which can be reduced to zero for some permutation $\boldsymbol{P}$, $G \subseteq G'$.

*Maximum common edge subgraph:* From Appendix B.2, we can write that $\text{GED}(G, G') = \min_{\boldsymbol{P}} 0.5(a^\oplus + a^\ominus) \left\|\max(\boldsymbol{A}, \boldsymbol{P}\boldsymbol{A}'\boldsymbol{P}^\top)\right\|_{1,1} + 0.5(b^\oplus + b^\ominus) \left\|\max\{\boldsymbol{\eta}_G, \boldsymbol{P}\boldsymbol{\eta}_{G'}\}\right\|_1 - a^\ominus|E'| - a^\oplus|E| - b^\ominus|V'| - b^\oplus|V|$. When $a^\ominus = a^\oplus = 1$ and $b^\oplus = b^\ominus = 0$, then $\text{GED}(G, G') = \left\|\max(\boldsymbol{A}, \boldsymbol{P}\boldsymbol{A}'\boldsymbol{P}^\top)\right\|_{1,1} = |E| + |E'| - \left\|\min(\boldsymbol{A}, \boldsymbol{P}\boldsymbol{A}'\boldsymbol{P}^\top)\right\|_{1,1}$. Here, $\min(\boldsymbol{A}, \boldsymbol{P}\boldsymbol{A}'\boldsymbol{P}^\top)$ characterizes maximum common edge subgraph and $\left\|\min(\boldsymbol{A}, \boldsymbol{P}\boldsymbol{A}'\boldsymbol{P}^\top)\right\|_{1,1}$ provides the number of edges of it.

### B.4 Relation between ALIGNDIFF and DIFFALIGN

**Lemma 2** *Let $\boldsymbol{Z}, \boldsymbol{Z}' \in \mathbb{R}^{N \times M}$, and $\boldsymbol{S} \in \mathbb{R}_{\geq 0}^{N \times N}$ be double stochastic. Then,*

$$\left\|\text{ReLU}\left(\boldsymbol{Z} - \boldsymbol{S}\boldsymbol{Z}'\right)\right\|_{1,1} \leq \sum_{i,j} \left\|\text{ReLU}\left(\boldsymbol{Z}[i,:] - \boldsymbol{Z}'[j,:]\right)\right\|_1 \boldsymbol{S}[i,j]$$

*Proof:* We can write,

$$\|\text{ReLU}\,(\boldsymbol{Z} - \boldsymbol{S}\boldsymbol{Z}')\|_{1,1} = \sum_{i,j} \left| \text{ReLU}\left(\boldsymbol{Z}[i,j] - \sum_k \boldsymbol{S}[i,k]\boldsymbol{Z}'[k,j]\right)\right|$$

$$\overset{(*)}{=} \sum_{i,j} \text{ReLU}\left(\sum_k \boldsymbol{S}[i,k]\boldsymbol{Z}[i,j] - \boldsymbol{S}[i,k]\boldsymbol{Z}'[k,j]\right)$$

$$\overset{(**)}{\leq} \sum_{i,j}\sum_k \boldsymbol{S}[i,k]\text{ReLU}\,(\boldsymbol{Z}[i,j] - \boldsymbol{Z}'[k,j])$$

$$= \sum_{i,k} \|\text{ReLU}\,(\boldsymbol{Z}[i,:] - \boldsymbol{Z}'[k,:])\|_1\, \boldsymbol{S}[i,k] \qquad \square$$

where $(*)$ follows since $\sum_k \boldsymbol{S}[i,k] = 1\,\forall i \in [N]$, and $(**)$ follows due to convexity of $\text{ReLU}\,()$. Now, notice that when $\boldsymbol{S} \in \mathbb{P}_N$, then $\boldsymbol{S}[i,:]$ is 1 at one element while 0 at the rest. In that case, we have

$$\sum_{i,j}\text{ReLU}\left(\sum_k \boldsymbol{S}[i,k]\boldsymbol{Z}[i,j] - \boldsymbol{S}[i,k]\boldsymbol{Z}'[k,j]\right) = \sum_{i,j}\text{ReLU}\,(\boldsymbol{Z}[i,j] - \boldsymbol{Z}'[k_i^*,j])$$

$$= \sum_{i,j}\sum_k \boldsymbol{S}[i,k]\text{ReLU}\,(\boldsymbol{Z}[i,j] - \boldsymbol{Z}'[k,j])$$

where $k_i^*$ is the index where $\boldsymbol{S}[i,:]$ is 1. Hence, we have an equality when $\boldsymbol{S}$ is a hard permutation. Replacing $(\boldsymbol{Z}, \boldsymbol{Z}')$ with $(\boldsymbol{R}, \boldsymbol{R}')$ and $(\boldsymbol{X}, \boldsymbol{X}')$, we get that ALIGNDIFF and DIFFALIGN are equivalent when $\boldsymbol{S}$ is a hard permutation matrix, and moreover DIFFALIGN is an upper bound on ALIGNDIFF when $\boldsymbol{S}$ is a soft permutation matrix.

### B.5  Proof that our design ensures conditions of Proposition 1

Here we show why it is necessary to have a symmetric form for $\boldsymbol{C}[u,u']$ in PERMNET$_\phi$.
For $\text{GED}(G, G')$,

$$\boldsymbol{C}[u,v] = \|c_\phi\,(\boldsymbol{x}_K(u)) - c_\phi\,(\boldsymbol{x}_K'(v))\|_1$$

For $\text{GED}(G', G)$,

$$\boldsymbol{C}'[v,u] = \|c_\phi\,(\boldsymbol{x}_K'(v)) - c_\phi\,(\boldsymbol{x}_K(u))\|_1$$

Because the Sinkhorn cost $\boldsymbol{C}[u,v]$ is symmetric, using the above equations we can infer,

$$\boldsymbol{C}[u,v] = \boldsymbol{C}'[v,u], \;\; \text{which implies} \;\; \boldsymbol{C}' = \boldsymbol{C}^\top$$

This leads to $\boldsymbol{P}' = \boldsymbol{P}^\top$. If we use an asymmetric Sinkhorn cost ($\boldsymbol{C}[u,v] = \|\text{ReLU}\,(c_\phi\,(\boldsymbol{x}_K(u)) - c_\phi\,(\boldsymbol{x}_K'(v)))\|_1$), we cannot ensure $\boldsymbol{C}[u,v] = \boldsymbol{C}'[v,u]$, which fails to satisfy $\boldsymbol{P} = \boldsymbol{P}^\top$.

### B.6  Alternative surrogate for GED

From Appendix B.2, we have

$$\text{GED}(G, G') = \min_{\boldsymbol{P} \in \mathbb{P}_N} \frac{a^\oplus + a^\ominus}{2}\left\|\max(\boldsymbol{A}, \boldsymbol{P}\boldsymbol{A}'\boldsymbol{P}^\top)\right\|_{1,1} - a^\ominus|E'| - a^\oplus|E|$$

$$+ \frac{b^\oplus + b^\ominus}{2}\left\|\max(\boldsymbol{\eta}_G, \boldsymbol{P}\boldsymbol{\eta}_{G'})\right\|_1 - b^\ominus|V'| - b^\oplus|V|$$

Following the relaxations done in Section 4.2, we propose an alternative neural surrogate by replacing $\left\|\max(\boldsymbol{A}, \boldsymbol{P}\boldsymbol{A}'\boldsymbol{P}^\top)\right\|_{1,1}$ by $\left\|\max(\boldsymbol{R}, \boldsymbol{S}\boldsymbol{R}')\right\|_{1,1}$ and $\left\|\max(\boldsymbol{\eta}_G, \boldsymbol{P}\boldsymbol{\eta}_{G'})\right\|_1$ by $\left\|\max(\boldsymbol{X}, \boldsymbol{P}\boldsymbol{X}')\right\|_{1,1}$, which gives us the approximated GED parameterized by $\theta$ and $\phi$ as

$$\text{GED}_{\theta,\phi}(G, G') = \frac{a^\oplus + a^\ominus}{2}\left\|\max(\boldsymbol{R}, \boldsymbol{S}\boldsymbol{R}')\right\|_{1,1} - a^\ominus|E'| - a^\oplus|E| \tag{20}$$

$$+ \frac{b^\oplus + b^\ominus}{2}\left\|\max(\boldsymbol{X}, \boldsymbol{P}\boldsymbol{X}')\right\|_{1,1} - b^\ominus|V'| - b^\oplus|V|$$

We call this neural surrogate as MAX. We note that element-wise maximum over $\boldsymbol{A}$ and $\boldsymbol{P}\boldsymbol{A}'\boldsymbol{P}^\top$, only allows non-edge to non-edge mapping attribute a value of zero. However, the neural surrogate described in Equation 20 fails to capture this, due to the presence of the soft alignment matrix $\boldsymbol{S}$. To address this, we explicitly discard such pairs from MAX by applying an OR operator over the edge presence between concerned node pairs, derived from the adjacency matrices $\boldsymbol{A}$ and $\boldsymbol{A}'$ and

populated in $\text{OR}(\boldsymbol{A}, \boldsymbol{A'}) \in \mathbb{R}^{\binom{N}{2} \times \binom{N}{2}}$ given by $\text{OR}(\boldsymbol{A}[u,v], \boldsymbol{A'}[u',v'])$. Similarly, the indication of node presence can be given be given as $\text{OR}(\boldsymbol{\eta}_G, \boldsymbol{\eta}_{G'})[u,u'] = \text{OR}(\boldsymbol{\eta}_G[u], \boldsymbol{\eta}_{G'}[u'])$. Hence, we write

$$\text{GED}_{\theta,\phi}(G, G') = \frac{a^{\oplus} + a^{\ominus}}{2} \left\| \text{OR}(\boldsymbol{A}, \boldsymbol{A'}) \odot \max(\boldsymbol{R}, \boldsymbol{S}\boldsymbol{R'}) \right\|_{1,1} - a^{\ominus} |E'| - a^{\oplus} |E|$$
$$+ \frac{b^{\oplus} + b^{\ominus}}{2} \left\| \text{OR}(\boldsymbol{\eta}_G, \boldsymbol{\eta}_{G'}) \odot \max(\boldsymbol{X}, \boldsymbol{P}\boldsymbol{X'}) \right\|_{1,1} - b^{\ominus} |V'| - b^{\oplus} |V|$$

$$(21)$$

We call this formulation as MAX-OR. We provide the comparison between MAX, MAX-OR, and our models in Appendix D.

## C  Details about experimental setup

### C.1  Generation of datasets

We have evaluated the performance of our methods and baselines on seven real-world datasets: Mutagenicity (Mutag), Ogbg-Code2 (Code2), Ogbg-Molhiv (Molhiv), Ogbg-Molpcba (Molpcba), AIDS, Linux and Yeast. We split each dataset into training, validation, and test splits in ratio of 60:20:20. For each split $\mathcal{D}$, we construct $(|\mathcal{D}|(|\mathcal{D}|+1))/2$ source and target graph instance pairs as follows: $\mathcal{S} = \{(G_i, G_j) : G_i, G_j \in \mathcal{D} \wedge i \leq j\}$. We perform experiment in *four* GED regimes:

1. GED under uniform cost functions, where $b^{\ominus} = b^{\oplus} = a^{\ominus} = a^{\oplus} = 1$ and substitution costs are 0
2. GED under non-uniform cost functions, where $b^{\ominus} = 3, b^{\oplus} = 1, a^{\ominus} = 2, a^{\oplus} = 1$ and substitution costs are 0
3. edge GED under non-uniform cost functions, where $b^{\ominus} = b^{\oplus} = 0$, $a^{\ominus} = 2, a^{\oplus} = 1$, and substitution costs are 0
4. GED with node substitution under uniform cost functions, where $b^{\ominus} = b^{\oplus} = a^{\ominus} = a^{\oplus} = 1$, as well as the node substitution cost $b^{\sim} = 1$.

We emphasize that we generated clean datasets by filtering out isomorphic graphs from the original datasets before performing the training, validation, and test splits. This step is crucial to prevent isomorphism bias in the models, which can occur due to leakage between the training and testing splits, as highlighted by [26].

For each graph, we have limited the maximum number of nodes to twenty, except for Linux, where the limit is ten. Information about the datasets is summarized in Table 7. Mutag contains nitroaromatic compounds, with each node having labels representing atom types. Molhiv and Molpcba contain molecules with node features representing atomic number, chirality, and other atomic properties. Code2 contains abstract syntax trees generated from Python codes. AIDS contains graphs of chemical compounds, with node types representing different atoms. For Molhiv, Molpcba and Linux, we have randomly sampled 1,000 graphs from each original dataset.

| | #Graphs | # Train Pairs | # Val Pairs | # Test Pairs | Avg. $|V|$ | Avg. $|E|$ | Avg. GED uniform cost | Avg. GED non-uniform cost |
|---|---|---|---|---|---|---|---|---|
| Mutag | 729 | 95703 | 10585 | 10878 | 16.01 | 15.76 | 11.15 | 18.57 |
| Code2 | 128 | 2926 | 325 | 378 | 18.77 | 17.77 | 10.02 | 16.43 |
| Molhiv | 1000 | 180300 | 20100 | 20100 | 15.01 | 15.65 | 11.77 | 19.86 |
| Molpcba | 1000 | 180300 | 20100 | 20100 | 17.52 | 18.67 | 9.58 | 15.73 |
| AIDS | 911 | 149331 | 16653 | 16836 | 10.97 | 10.97 | 7.38 | 12.07 |
| Yeast | 1000 | 180300 | 20100 | 20100 | 16.59 | 17.04 | 10.65 | 17.74 |
| Linux | 89 | 1431 | 153 | 190 | 8.71 | 8.35 | 4.91 | 7.94 |

Table 7: Salient characteristics of data sets.

### C.2  Details about state-of-the-art baselines

We compared our model against nine state-of-the-art neural baselines and three combinatorial GED baselines. Below, we provide details of the methodology and hyperparameter settings used for each baseline. We ensured that the number of model parameters were in a comparable range. Specifically, we set the number of GNN layers to 5, each with a node embedding dimension of 10, to ensure consistency and comparability with our model. The following hyperparameters are used for training: Adam optimiser with a learning rate of 0.001 and weight decay of 0.0005, batch size of 256, early stopping with patience of 100 epochs, and Sinkhorn temperature set to 0.01. **Neural Baselines:**

- **GMN-Match and GMN-Embed**  Graph Matching Networks (GMN) use Euclidean distance to assess the similarity between graph-level embeddings of each graph. GMN is available in two variants: GMN-Embed, a late interaction model, and GMN-Match, an early interaction model. For this study, we used the official implementation of GMN to compute Graph Edit Distance (GED).[1]
- **ISONET**  ISONET utilizes the Gumbel-Sinkhorn operator to learn asymmetric edge alignments between two graphs for subgraph matching. In our study, we extend ISONET's approach to predict the Graph Edit Distance (GED) score. We utilized the official PyTorch implementation provided by the authors for our experiments.[2]

---

[1] https://github.com/Lin-Yijie/Graph-Matching-Networks/tree/main
[2] https://github.com/Indradyumna/ISONET

- **GREED** GREED utilizes a siamese network architecture to compute graph-level embeddings in parallel for two graphs. It calculates the Graph Edit Distance (GED) score by computing the norm of the difference between these embeddings. The official implementation provided by the authors was used for our experiments.[3]
- **ERIC** ERIC utilizes a regularizer to learn node alignment, eliminating the need for an explicit node alignment module. The similarity score is computed using a Neural Tensor Network (NTN) and a Multi-Layer Perceptron (MLP) applied to the final graph-level embeddings of both graphs. These embeddings are derived by concatenating graph-level embeddings from each layer of a Graph Isomorphism Network (GIN). The model is trained using a combined loss from the regularizer and the predicted similarity score. For our experiments, we used the official PyTorch implementation to compute the Graph Edit Distance (GED). The GED scores were inverse normalized from the model output to predict the absolute GED.[4]
- **SimGNN** SimGNN leverages both graph-level and node-level embeddings at each layer of the GNN. The graph-level embeddings are processed through a Neural Tensor Network to obtain a pair-level embedding. Concurrently, the node-level embeddings are used to compute a pairwise similarity matrix between nodes, which is then converted into a histogram feature vector. A similarity score is calculated by passing the concatenation of these embeddings through a Multi-Layer Perceptron (MLP). We used the official PyTorch implementation of SimGNN and inverse normalization of the predicted Graph Edit Distance (GED) score to obtain the absolute GED value.[5]
- **H2MN** H2MN presents an early interaction model for graph similarity tasks. Instead of learning pairwise node relations, this method attempts to find higher-order node similarity using hypergraphs. At each time step of the hypergraph convolution, a subgraph matching module is employed to learn cross-graph similarity. After the convolution layers, a readout function is utilized to obtain graph-level embeddings. These embeddings are then concatenated and passed through a Multi-Layer Perceptron (MLP) to compute the similarity score. We used the official PyTorch implementation of H2MN.[6]
- **GraphSim** GraphSim uses GNN, where at each layer, a node-to-node similarity matrix is computed using the node embeddings. These similarity matrices are then processed using Convolutional Neural Networks (CNNs) and Multi-Layer Perceptrons (MLPs) to calculate a similarity score. We utilized the official PyTorch implementation.[7]
- **EGSC** We used the Teacher model proposed by Efficient Graph Similarity Computation (EGSC), which leverages an Embedding Fusion Network (EFN) at each layer of the Graph Isomorphism Network (GIN). The EFN generates a single embedding from a pair of graph embeddings. The embeddings of the graph pair from each layer are concatenated and subsequently passed through an additional EFN layer and a Multi-Layer Perceptron (MLP) to obtain the similarity score. To predict the absolute Graph Edit Distance (GED), we inversely normalized the GED score obtained from the output of EGSC. We utilized the official PyTorch implementation provided by the authors for our experiments. [8]

**Combinatorial Baselines:** We use the GEDLIB[9] library for implementation of all combinatorial baselines.

- **Bipartite** [41] Bipartite is an approximate algorithm that considers nodes and surrounding edges of nodes into account try to make a bipartite matching between two graphs. They use linear assignment algorithms to match nodes and their surroundings in two graphs.
- **Branch [8], Branch Tight [8]** improve upon [41] by decomposing graphs into branches. Branch Tight algorithm is another version of Branch that calculates a tighter lower bound but has a higher time complexity than Branch.
- **Anchor Aware GED** Chang et al. [15] provides an approximation algorithm that calculates a tighter lower bound using the anchor aware technique.

---

[3] https://github.com/idea-iitd/greed
[4] https://github.com/JhuoW/ERIC
[5] https://github.com/benedekrozemberczki/SimGNN
[6] https://github.com/cszhangzhen/H2MN
[7] https://github.com/yunshengb/GraphSim
[8] https://github.com/canqin001/Efficient_Graph_Similarity_Computation
[9] https://github.com/dblumenthal/gedlib

- **IPFP** [11] is an approximation algorithm which handles node and edge mapping simultaneously unlike previously discussed methods. This solves a quadratic assignment problem on edges and nodes.
- **F2** [29] uses a binary linear programming approach to find a higher lower bound on GED calculation. This method was used with a very high time limit to generate Ground truth for our experiments.

### C.3 Details about GRAPHEDX

At the high level, GRAPHEDX consists of two components $\text{EMBED}_\theta$ and $\text{PERMNET}_\phi$.

**Neural Parameterization of $\text{EMBED}_\theta$:** $\text{EMBED}_\theta$ consists of two modules: a GNN denoted as $\text{MPNN}_\theta$ and a $\text{MLP}_\theta$. The $\text{MPNN}_\theta$ consists of $K = 5$ propagation layers used to compute node embeddings of dimension $d = 10$. At each layer $k$, we compute the updated the node embedding as follows:

$$\boldsymbol{x}_{k+1}(u) = \text{UPDATE}_\theta \left( \boldsymbol{x}_k(u), \sum_{v \in \text{nbr}(u)} \text{LRL}_\theta(\boldsymbol{x}_k(u), \boldsymbol{x}_k(v)) \right) \tag{22}$$

where $\text{LRL}_\theta$ is a Linear-ReLU-Linear network, with $d = 10$ features, and the $\text{UPDATE}_\theta$ network consists of a Gated Recurrent Unit [30]. In case of GED setting under uniform cost and GED setting under non-uniform cost, we set the initial node features $\boldsymbol{x}_0(u) = 1$, following [30]. However, in case of computation of GED with node substitution costs, we explicitly provide the one-hot labels as node features. Given the node embeddings and edge-presence indicator obtained from the adjacency matrices, after 5 layer propogations, we compute the edge embeddings $\boldsymbol{r}(e)$ using $\text{MLP}_\theta$, which is decoupled from $\text{MPNN}_\theta$. $\text{MLP}_\theta$ consists of a Linear-ReLU-Linear network that maps the $2d + 1 = 21$ dimensional input consisting of forward $(\boldsymbol{x}_K(u) \,\|\, \boldsymbol{x}_K(v) \,\|\, \boldsymbol{A}[u, v])$ and backward $(\boldsymbol{x}_K(v) \,\|\, \boldsymbol{x}_K(u) \,\|\, \boldsymbol{A}[v, u])$ signals to $D = 20$ dimensions.

**Neural Parameterization of $\text{PERMNET}_\phi$:** Given the node embeddings $\boldsymbol{x}_K(\cdot)$ and $\boldsymbol{x}'_K(\cdot)$, we first pass them through a neural network $c_\phi$ which consists of a Linear-ReLU-Linear network transforming the features from $d = 10$ to $N$ dimensions, which is the number of nodes after padding. Except for Linux where $N = 10$, all other datasets have $N = 20$. We obtain the matrix $\boldsymbol{C}$ such that $\boldsymbol{C}[u, u'] = \|c_\phi(\boldsymbol{x}_K(u)) - c_\phi(\boldsymbol{x}'_K(u'))\|_1$. Using temperature $\tau = 0.01$, we perform Sinkhorn iterations on $\exp(-\boldsymbol{C}/\tau)$ as follows for $T = 20$ iterations to get $\boldsymbol{P}$:

$$\boldsymbol{P}_k = \text{NORMCOL}\left(\text{NORMROW}\left(\boldsymbol{P}_{k-1}\right)\right)$$

where $\boldsymbol{P}_0 = \exp(-\boldsymbol{C}/\tau)$. Here $\text{NORMROW}(\boldsymbol{M})[i, j] = \boldsymbol{M}[i, j]/\sum_\ell \boldsymbol{M}[\ell, j]$ denotes the row normalization function and $\text{NORMCOL}(\boldsymbol{M})[i, j] = \boldsymbol{M}[i, j]/\sum_\ell \boldsymbol{M}[i, \ell]$ denotes the column normalization function. We note that the soft alignment $\boldsymbol{P}$ obtained does not depend on the GED cost values, as discussed in Appendix B. The soft alignment $\boldsymbol{P}$ for nodes is used to construct soft alignment $\boldsymbol{S}$ for as follows: $\boldsymbol{S}[(u, v), (u', v')] = \boldsymbol{P}[u, u']\boldsymbol{P}[v, v'] + \boldsymbol{P}[u, v']\boldsymbol{P}[v, u']$.

### C.4 Evaluation metrics

Given the dataset $\mathcal{S}$ consisting of input pairs of graphs $(G, G')$ along with the ground truth $\text{GED}(G, G')$ and model prediction $\widehat{\text{GED}}(G, G')$, we evaluate the performance of the model using the Root Mean Square Error (RMSE) and Kendall-Tau (KTau) [28] between the predicted GED scores and actual GED values.

- **MSE:** It evaluates how far the predicted GED values are from the ground truth. A better performing model is indicated by a lower MSE value.

$$\text{MSE} = \frac{1}{|\mathcal{S}|} \sum_{(G, G') \in \mathcal{S}} \left( \text{GED}(G, G') - \widehat{\text{GED}}(G, G') \right)^2 \tag{23}$$

- **KTau:** Selection of relevant corpus graphs via graph similarity scoring is crucial to graph retrieval setups. In this context, we would like the number of concordant pairs $N_+$ (where the ranking of ground truth GED and model prediction agree) to be high, and the discordant pairs $N_-$ (where the two disagree) to be low. Formally, we write

$$\text{KTau} = \frac{N_+ - N_-}{\binom{|\mathcal{S}|}{2}} \tag{24}$$

For the methods which compute a similarity score between the pair of graphs through the notion of normalized GED, we map the similarity score $s$ back to the GED as $\widehat{\text{GED}}(G, G') = -\frac{|V| + |V|'}{2} \log(s + \epsilon)$ where $\epsilon = 10^{-7}$ is added for stability of the logarithm.

### C.5 Hardware and license

We implement our models using Python 3.11.2 and PyTorch 2.0.0. The training of our models and the baselines was performed across servers containing Intel Xeon Silver 4216 2.10GHz CPUs, and Nvidia RTX A6000 GPUs. Running times of all methods are compared on the same GPU.

# D  Additional experiments

In this section, we present results from various additional experiments performed to measure the performance of our model under different cost settings.

## D.1  Comparison of performance of GRAPHEDX on non-uniform cost Edge-GED

We consider another cost setting – where the node costs are explicitly set to 0, and $a^\oplus = 1, a^\ominus = 2$. In such a case, GRAPHEDX only consists of $\Delta^\ominus(\boldsymbol{R}, \boldsymbol{R'} \,|\, \boldsymbol{S})$ and $\Delta^\oplus(\boldsymbol{R}, \boldsymbol{R'} \,|\, \boldsymbol{S})$ terms. To showcase the importance of aligning edges through edge alignment, we generate an alternate model, where the alignment happens through the terms $\Delta^\ominus(\boldsymbol{X}, \boldsymbol{X'} \,|\, \boldsymbol{P})$ and $\Delta^\oplus(\boldsymbol{X}, \boldsymbol{X'} \,|\, \boldsymbol{P})$, where we set $b^\oplus = 1$ and $b^\ominus = 2$, and set the edge costs to 0. We call this model NodeSwap (w/o XOR), and the corresponding XOR variant as NodeSwap + XOR. In Table 8, we compare the performance variants of GRAPHEDX with NodeSwap (w/o XOR) and the rest of the baselines to predict the Edge GED score in an non-uniform cost setting. From the results, we can infer that the performance of edge-alignment based model to predict Edge-GED outperforms the corresponding node-alignment version.

| | MSE $\pm$ STD | | | KTau | | |
| --- | --- | --- | --- | --- | --- | --- |
| | Mutag | Molhiv | Linux | Mutag | Molhiv | Linux |
| GMN-Match | $11.276 \pm 0.143$ | $13.586 \pm 0.171$ | $4.893 \pm 0.527$ | 0.600 | 0.562 | 0.453 |
| GMN-Embed | $13.627 \pm 0.179$ | $16.482 \pm 0.188$ | $4.363 \pm 0.420$ | 0.556 | 0.529 | 0.484 |
| ISONET | $1.468 \pm 0.020$ | $2.142 \pm 0.023$ | $1.930 \pm 0.186$ | 0.846 | 0.802 | 0.659 |
| GREED | $11.906 \pm 0.148$ | $13.723 \pm 0.136$ | $3.847 \pm 0.397$ | 0.588 | 0.558 | 0.512 |
| ERIC | $1.900 \pm 0.028$ | $2.154 \pm 0.024$ | $3.361 \pm 0.353$ | 0.823 | 0.805 | 0.510 |
| SimGNN | $3.138 \pm 0.052$ | $3.771 \pm 0.046$ | $5.089 \pm 0.524$ | 0.784 | 0.736 | 0.410 |
| H2MN | $3.771 \pm 0.062$ | $3.735 \pm 0.047$ | $5.443 \pm 0.566$ | 0.748 | 0.741 | 0.358 |
| GraphSim | $4.696 \pm 0.076$ | $5.200 \pm 0.074$ | $6.597 \pm 0.697$ | 0.720 | 0.694 | 0.316 |
| EGSC | $1.871 \pm 0.028$ | $2.187 \pm 0.025$ | $2.803 \pm 0.260$ | 0.823 | 0.797 | 0.608 |
| NodeSwap (w/o XOR) | $1.246 \pm 0.017$ | $1.858 \pm 0.019$ | $0.997 \pm 0.124$ | 0.857 | 0.814 | 0.757 |
| NodeSwap + XOR | $11.984 \pm 0.227$ | $11.158 \pm 0.196$ | $10.959 \pm 1.116$ | 0.586 | 0.604 | 0.321 |
| GRAPHEDX (w/o XOR) | $1.174 \pm 0.016$ | $1.842 \pm 0.019$ | $0.976 \pm 0.115$ | 0.863 | 0.815 | 0.764 |
| GRAPHEDX + XOR | $1.125 \pm 0.016$ | $1.855 \pm 0.020$ | $0.922 \pm 0.108$ | 0.866 | 0.817 | 0.780 |

Table 8: Comparison of edge-alignment based GED scoring function with node-alignment based GED scoring function and state-of-the-art baselines under the cost setting: $a^\ominus = 2, a^\oplus = 1, b^\ominus = b^\oplus = 0$. In case of NodeSwap (w/o XOR), we swap the edge costs and node costs, and expect the model to learn the alignments in Edge GED through node alignment only. Green (yellow) numbers report the best (second best) performers.

## D.2 Comparison of GRAPHEDX with baselines on uniform and non-uniform cost setting

Tables 9 and 10 report performance in terms of MSE under uniform and non-uniform cost settings, respectively. Table 11 reports performance in terms of KTau under both uniform and non-uniform cost settings. The results are similar to those in Table 3, where our model is the clear winner across all datasets, outperforming the second-best performer by a significant margin. There is no consistent second-best model, but ERIC, EGSC, and ISONET perform comparably and better than the others.

| | Mutag | Code2 | Molhiv | Molpcba | AIDS | Linux | Yeast |
|---|---|---|---|---|---|---|---|
| GMN-Match | $0.797 \pm 0.013$ | $1.677 \pm 0.187$ | $1.318 \pm 0.020$ | $1.073 \pm 0.011$ | $0.821 \pm 0.010$ | $0.687 \pm 0.088$ | $1.175 \pm 0.013$ |
| GMN-Embed | $1.032 \pm 0.016$ | $1.358 \pm 0.104$ | $1.859 \pm 0.020$ | $1.951 \pm 0.020$ | $1.044 \pm 0.013$ | $0.736 \pm 0.102$ | $1.767 \pm 0.021$ |
| ISONET | $1.187 \pm 0.021$ | $0.879 \pm 0.061$ | $1.354 \pm 0.015$ | $1.106 \pm 0.011$ | $1.640 \pm 0.020$ | $1.185 \pm 0.115$ | $1.578 \pm 0.019$ |
| GREED | $1.398 \pm 0.033$ | $1.869 \pm 0.140$ | $1.708 \pm 0.019$ | $1.550 \pm 0.017$ | $1.004 \pm 0.012$ | $1.331 \pm 0.169$ | $1.423 \pm 0.015$ |
| ERIC | $0.719 \pm 0.011$ | $1.363 \pm 0.110$ | $1.165 \pm 0.018$ | $0.862 \pm 0.009$ | $0.731 \pm 0.008$ | $1.664 \pm 0.260$ | $0.969 \pm 0.010$ |
| SimGNN | $1.471 \pm 0.024$ | $2.667 \pm 0.215$ | $1.609 \pm 0.020$ | $1.456 \pm 0.020$ | $1.455 \pm 0.020$ | $7.232 \pm 0.762$ | $1.999 \pm 0.043$ |
| H2MN | $1.278 \pm 0.021$ | $7.240 \pm 0.527$ | $1.521 \pm 0.020$ | $1.402 \pm 0.020$ | $1.114 \pm 0.015$ | $2.238 \pm 0.247$ | $1.353 \pm 0.018$ |
| GraphSim | $2.005 \pm 0.031$ | $3.139 \pm 0.206$ | $2.577 \pm 0.064$ | $1.656 \pm 0.023$ | $1.936 \pm 0.026$ | $2.900 \pm 0.318$ | $2.232 \pm 0.030$ |
| EGSC | $0.765 \pm 0.011$ | $4.165 \pm 0.285$ | $1.138 \pm 0.016$ | $0.938 \pm 0.010$ | $0.627 \pm 0.007$ | $2.411 \pm 0.325$ | $0.950 \pm 0.010$ |
| GRAPHEDX | $0.492 \pm 0.007$ | $0.429 \pm 0.036$ | $0.781 \pm 0.008$ | $0.764 \pm 0.007$ | $0.565 \pm 0.006$ | $0.354 \pm 0.043$ | $0.717 \pm 0.007$ |

Table 9: Comparison with baselines in terms of MSE including standard error for uniform cost setting ($b^{\ominus} = b^{\oplus} = a^{\ominus} = a^{\oplus} = 1$). Green (yellow) numbers report the best (second best) performers.

| | Mutag | Code2 | Molhiv | Molpcba | AIDS | Linux | Yeast |
|---|---|---|---|---|---|---|---|
| GMN-Match | $69.210 \pm 0.883$ | $13.472 \pm 0.970$ | $76.923 \pm 0.862$ | $23.985 \pm 0.224$ | $31.522 \pm 0.513$ | $21.519 \pm 2.256$ | $63.179 \pm 1.127$ |
| GMN-Embed | $72.495 \pm 0.915$ | $13.425 \pm 1.035$ | $78.254 \pm 0.865$ | $28.437 \pm 0.268$ | $33.221 \pm 0.523$ | $20.591 \pm 2.136$ | $60.949 \pm 0.663$ |
| ISONET | $3.369 \pm 0.062$ | $3.025 \pm 0.206$ | $3.451 \pm 0.039$ | $2.781 \pm 0.029$ | $5.513 \pm 0.092$ | $3.031 \pm 0.299$ | $4.555 \pm 0.061$ |
| GREED | $68.732 \pm 0.867$ | $11.095 \pm 0.773$ | $78.300 \pm 0.795$ | $26.057 \pm 0.238$ | $34.354 \pm 0.557$ | $20.667 \pm 2.140$ | $60.652 \pm 0.704$ |
| ERIC | $1.981 \pm 0.032$ | $12.767 \pm 1.177$ | $3.377 \pm 0.070$ | $2.057 \pm 0.020$ | $1.581 \pm 0.017$ | $7.809 \pm 0.911$ | $2.341 \pm 0.030$ |
| SimGNN | $4.747 \pm 0.079$ | $5.212 \pm 0.360$ | $4.145 \pm 0.051$ | $3.465 \pm 0.047$ | $4.316 \pm 0.071$ | $5.369 \pm 0.546$ | $4.496 \pm 0.060$ |
| H2MN | $3.413 \pm 0.053$ | $9.435 \pm 0.728$ | $3.782 \pm 0.046$ | $3.396 \pm 0.046$ | $3.105 \pm 0.043$ | $5.848 \pm 0.611$ | $3.678 \pm 0.046$ |
| GraphSim | $5.370 \pm 0.092$ | $7.405 \pm 0.577$ | $6.643 \pm 0.181$ | $3.928 \pm 0.053$ | $5.266 \pm 0.081$ | $6.815 \pm 0.628$ | $6.907 \pm 0.137$ |
| EGSC | $1.758 \pm 0.026$ | $3.957 \pm 0.365$ | $2.371 \pm 0.025$ | $2.133 \pm 0.022$ | $1.693 \pm 0.023$ | $5.503 \pm 0.496$ | $2.157 \pm 0.027$ |
| GRAPHEDX | $1.134 \pm 0.016$ | $1.478 \pm 0.118$ | $1.804 \pm 0.019$ | $1.677 \pm 0.016$ | $1.252 \pm 0.014$ | $0.914 \pm 0.110$ | $1.603 \pm 0.016$ |

Table 10: Comparison with baselines in terms of MSE including standard error for non-uniform cost setting ($b^{\ominus} = 3, b^{\oplus} = 1, a^{\ominus} = 2, a^{\oplus} = 1$). Green (yellow) numbers report the best (second best) performers.

| | GED with uniform cost | | | | | | | non-uniform cost | | | | | | |
|---|---|---|---|---|---|---|---|---|---|---|---|---|---|---|
| | Mutag | Code2 | Molhiv | Molpcba | AIDS | Linux | Yeast | Mutag | Code2 | Molhiv | Molpcba | AIDS | Linux | Yeast |
| GMN-Match | 0.901 | 0.876 | 0.887 | 0.797 | 0.824 | 0.826 | 0.852 | 0.606 | 0.781 | 0.619 | 0.596 | 0.611 | 0.438 | 0.610 |
| GMN-Embed | 0.887 | 0.892 | 0.856 | 0.723 | 0.796 | 0.815 | 0.815 | 0.603 | 0.790 | 0.607 | 0.534 | 0.601 | 0.531 | 0.573 |
| ISONET | 0.885 | 0.918 | 0.878 | 0.793 | 0.756 | 0.786 | 0.827 | 0.887 | 0.908 | 0.875 | 0.817 | 0.755 | 0.776 | 0.834 |
| GREED | 0.873 | 0.878 | 0.859 | 0.757 | 0.807 | 0.756 | 0.832 | 0.614 | 0.812 | 0.598 | 0.547 | 0.596 | 0.522 | 0.582 |
| ERIC | 0.909 | 0.892 | 0.897 | 0.820 | 0.837 | 0.736 | 0.868 | 0.620 | 0.804 | 0.895 | 0.841 | 0.855 | 0.633 | 0.886 |
| SimGNN | 0.871 | 0.856 | 0.877 | 0.776 | 0.775 | 0.377 | 0.834 | 0.862 | 0.874 | 0.872 | 0.804 | 0.768 | 0.731 | 0.843 |
| H2MN | 0.878 | 0.711 | 0.879 | 0.781 | 0.794 | 0.664 | 0.848 | 0.873 | 0.813 | 0.875 | 0.804 | 0.792 | 0.681 | 0.851 |
| GraphSim | 0.847 | 0.839 | 0.856 | 0.756 | 0.730 | 0.601 | 0.810 | 0.851 | 0.844 | 0.851 | 0.784 | 0.744 | 0.656 | 0.824 |
| EGSC | 0.906 | 0.815 | 0.896 | 0.809 | 0.850 | 0.664 | 0.868 | 0.912 | 0.894 | 0.900 | 0.836 | 0.858 | 0.696 | 0.884 |
| GRAPHEDX | 0.926 | 0.937 | 0.910 | 0.831 | 0.857 | 0.882 | 0.886 | 0.929 | 0.932 | 0.912 | 0.858 | 0.871 | 0.875 | 0.898 |

Table 11: Comparison with baselines in terms of KTau for both uniform and non-uniform cost settings, where for uniform cost settings costs are $b^{\ominus} = b^{\oplus} = a^{\ominus} = a^{\oplus} = 1$ and for non-uniform cost settings costs are $b^{\ominus} = 3, b^{\oplus} = 1, a^{\ominus} = 2, a^{\oplus} = 1$. Green (yellow) numbers report the best (second best) performers.

## D.3 Comparison of GRAPHEDX with baselines with and without cost features

Table 12 reports performance in terms of MSE under non-uniform cost setting, with and without costs used as features to the baselines. We notice that that in some cases, providing cost features boost the performance of baselines significantly, and in a few cases, withholding the costs gives a slight improvement in performance. However, GRAPHEDX, which uses costs in the distance formulation rather than features, outperforms all baselines by a significant margin.

| | Cost used as features | Mutag | Code2 | Molhiv | Molpcba | AIDS | Linux | Yeast |
|---|---|---|---|---|---|---|---|---|
| GMN-Match | ✓ | 69.210 | 13.472 | **76.923** | **23.985** | **31.522** | 21.519 | 63.179 |
| | ✗ | **68.635** | **12.769** | 84.113 | 24.471 | 31.636 | **20.255** | **62.715** |
| GMN-Embed | ✓ | **72.495** | **13.425** | **78.254** | **28.437** | **33.221** | 20.591 | 60.949 |
| | ✗ | 87.581 | 18.189 | 80.797 | 30.276 | 34.752 | **20.227** | **59.941** |
| ISONET | ✓ | **3.369** | 3.025 | **3.451** | 2.781 | **5.513** | 3.031 | **4.555** |
| | ✗ | 3.850 | 1.780 | 3.507 | 2.906 | 5.865 | 2.771 | 4.861 |
| GREED | ✓ | **68.732** | **11.095** | **78.300** | 26.057 | 34.354 | **20.667** | 60.652 |
| | ✗ | 78.878 | 12.774 | 78.837 | 26.188 | **32.318** | 23.478 | **55.985** |
| ERIC | ✓ | 1.981 | 12.767 | 3.377 | 2.057 | 1.581 | **7.809** | 2.341 |
| | ✗ | **1.912** | **12.391** | **2.588** | 2.220 | **1.536** | 11.186 | **2.161** |
| SimGNN | ✓ | 4.747 | **5.212** | 4.145 | 3.465 | 4.316 | **5.369** | 4.496 |
| | ✗ | **2.991** | 8.923 | **4.062** | **3.397** | **3.470** | 6.623 | **4.289** |
| H2MN | ✓ | 3.413 | **9.435** | 3.782 | 3.396 | 3.105 | 5.848 | **3.678** |
| | ✗ | **3.287** | 14.892 | **3.611** | **3.377** | **3.064** | **5.576** | 3.776 |
| GraphSim | ✓ | 5.370 | **7.405** | 6.643 | 3.928 | **5.266** | 6.815 | 6.907 |
| | ✗ | **4.886** | 10.257 | **6.394** | **3.921** | 5.538 | **6.439** | **6.033** |
| EGSC | ✓ | 1.758 | 3.957 | 2.371 | 2.133 | 1.693 | 5.503 | 2.157 |
| | ✗ | 1.769 | 4.395 | 2.510 | 2.217 | 1.432 | 4.664 | 2.305 |
| GRAPHEDX | ✗ | 1.134 | 1.478 | 1.804 | 1.677 | 1.252 | 0.914 | 1.603 |

Table 12: Comparison of performance (MSE) of methods for the non-uniform cost setting when nodes are initialized with costs as features versus without. For each method, the better performance between with and without cost-feature initialization is highlighted in bold for both uniform and non-uniform cost settings. In each column, Green (yellow) numbers report the best (second best) performers.

## D.4 Comparison of GRAPHEDX with baselines with node substitution cost

In Tables 13 and 14, we compare the performance of GRAPHEDX with baselines under a node substitution cost $b^\sim$. The cost setting is $b^\ominus = b^\oplus = b^\sim = a^\ominus = a^\oplus = 1$. This experiment includes only five datasets where node labels are present. We observe that GRAPHEDX outperforms all other baselines. There is no clear second-best model, but ERIC, EGSC, and ISONET perform better than the others.

| | Mutag | Code2 | Molhiv | Molpcba | AIDS |
|---|---|---|---|---|---|
| GMN-Match | $1.057 \pm 0.011$ | $5.224 \pm 0.404$ | $1.388 \pm 0.018$ | $1.432 \pm 0.017$ | $0.868 \pm 0.007$ |
| GMN-Embed | $2.159 \pm 0.026$ | $4.070 \pm 0.318$ | $3.523 \pm 0.040$ | $4.657 \pm 0.054$ | $1.818 \pm 0.014$ |
| ISONET | $0.876 \pm 0.008$ | $1.129 \pm 0.084$ | $1.617 \pm 0.020$ | $1.332 \pm 0.014$ | $1.142 \pm 0.010$ |
| GREED | $2.876 \pm 0.032$ | $4.983 \pm 0.531$ | $2.923 \pm 0.033$ | $3.902 \pm 0.044$ | $2.175 \pm 0.016$ |
| ERIC | $0.886 \pm 0.009$ | $6.323 \pm 0.683$ | $1.537 \pm 0.018$ | $1.278 \pm 0.014$ | $1.602 \pm 0.036$ |
| SimGNN | $1.160 \pm 0.013$ | $5.909 \pm 0.490$ | $1.888 \pm 0.031$ | $2.172 \pm 0.050$ | $1.418 \pm 0.020$ |
| H2MN | $1.277 \pm 0.014$ | $6.783 \pm 0.587$ | $1.891 \pm 0.024$ | $1.666 \pm 0.021$ | $1.290 \pm 0.011$ |
| GraphSim | $1.043 \pm 0.010$ | $4.708 \pm 0.425$ | $1.817 \pm 0.021$ | $1.748 \pm 0.021$ | $1.561 \pm 0.021$ |
| EGSC | $0.776 \pm 0.008$ | $8.742 \pm 0.831$ | $1.273 \pm 0.016$ | $1.426 \pm 0.018$ | $1.270 \pm 0.028$ |
| GRAPHEDX | $0.441 \pm 0.004$ | $0.820 \pm 0.092$ | $0.792 \pm 0.009$ | $0.846 \pm 0.009$ | $0.538 \pm 0.003$ |

Table 13: Comparison with baselines in terms of MSE including standard error, in presence of the node substitution cost, which set to one in uniform cost setting: $b^\ominus = b^\oplus = b^\sim = a^\ominus = a^\oplus = 1$. Green (yellow) numbers report the best (second best) performers.

| | Mutag | Code2 | Molhiv | Molpcba | AIDS |
|---|---|---|---|---|---|
| GMN-Match | 0.895 | 0.811 | 0.881 | 0.809 | 0.839 |
| GMN-Embed | 0.847 | 0.845 | 0.796 | 0.684 | 0.767 |
| ISONET | 0.906 | 0.925 | 0.868 | 0.815 | 0.812 |
| GREED | 0.827 | 0.829 | 0.822 | 0.710 | 0.746 |
| ERIC | 0.905 | 0.847 | 0.872 | 0.818 | 0.815 |
| SimGNN | 0.891 | 0.836 | 0.864 | 0.797 | 0.810 |
| H2MN | 0.886 | 0.818 | 0.858 | 0.789 | 0.802 |
| GraphSim | 0.896 | 0.846 | 0.860 | 0.782 | 0.795 |
| EGSC | 0.912 | 0.802 | 0.885 | 0.821 | 0.832 |
| GRAPHEDX | 0.936 | 0.945 | 0.913 | 0.856 | 0.874 |

Table 14: Comparison with baselines in terms of KTau, in presence of the node substitution cost, which set to one in uniform cost setting: $b^\ominus = b^\oplus = b^\sim = a^\ominus = a^\oplus = 1$. Green (yellow) numbers report the best (second best) performers.

## D.5 Performance evaluation for edge-only vs. all-node-pair representations

In this section, we compare the performance of using graph representation with two variants of our method. (i) Edge-only (edge $\rightarrow$ edge): Here, $\boldsymbol{R}, \boldsymbol{R'} \in \mathbb{R}^{\max(|E|,|E'|) \times D}$ are computed using only the embeddings of node-pairs that are edges, and excluding non-edges. This means that $\boldsymbol{S}$ becomes an edge-to-edge alignment matrix instead of a full node-pair alignment matrix. (ii) Edge-only (pair $\rightarrow$ pair): In this variant, $\boldsymbol{S}$ remains a node-pair alignment matrix, but the embeddings of the non-edges in $\boldsymbol{R}, \boldsymbol{R'} \in \mathbb{R}^{N(N-1)/2 \times D}$ are explicitly set to zero. Tables 15 and 16 contain extended results from Table 6 across seven datasets. The results are similar to those discussed in the main paper: (1) both these sparse representations perform significantly worse compared to our method using non-trivial representations for both edges and non-edges, and (2) Edge-only (edge $\rightarrow$ edge) performs better than Edge-only (pair $\rightarrow$ pair). This underscores the importance of explicitly modeling trainable non-edge embeddings to capture the sensitivity of GED to global graph structure.

| | Mutag | Code2 | Molhiv | Molpcba | AIDS | Linux | Yeast |
|---|---|---|---|---|---|---|---|
| Edge-only (edge $\rightarrow$ edge) | $0.566 \pm 0.008$ | $0.683 \pm 0.051$ | $0.858 \pm 0.009$ | $0.791 \pm 0.008$ | $0.598 \pm 0.006$ | $0.454 \pm 0.063$ | $0.749 \pm 0.007$ |
| Edge-only (pair $\rightarrow$ pair) | $0.596 \pm 0.008$ | $0.760 \pm 0.058$ | $0.862 \pm 0.009$ | $0.811 \pm 0.008$ | $0.606 \pm 0.006$ | $0.474 \pm 0.056$ | $0.761 \pm 0.008$ |
| GRAPHEDX | $0.492 \pm 0.007$ | $0.429 \pm 0.036$ | $0.781 \pm 0.008$ | $0.764 \pm 0.007$ | $0.565 \pm 0.006$ | $0.354 \pm 0.043$ | $0.717 \pm 0.007$ |

Table 15: Comparison of using all-node-pairs against edge-only representations using MSE for uniform cost setting. Green (yellow) numbers report the best (second best) performers.

| | Mutag | Code2 | Molhiv | Molpcba | AIDS | Linux | Yeast |
|---|---|---|---|---|---|---|---|
| Edge-only (edge → edge) | 1.274 ± 0.017 | 1.817 ± 0.141 | 1.847 ± 0.019 | 1.793 ± 0.017 | 1.318 ± 0.014 | 0.907 ± 0.129 | 1.649 ± 0.016 |
| Edge-only (pair → pair) | 1.276 ± 0.017 | 1.879 ± 0.136 | 1.865 ± 0.020 | 1.779 ± 0.017 | 1.422 ± 0.015 | 0.992 ± 0.114 | 1.694 ± 0.017 |
| GRAPHEDX | 1.134 ± 0.016 | 1.478 ± 0.118 | 1.804 ± 0.019 | 1.677 ± 0.016 | 1.252 ± 0.014 | 0.914 ± 0.110 | 1.603 ± 0.016 |

Table 16: Comparison of using all-node-pairs against edge-only representations using MSE for non-uniform cost setting. Green (yellow) numbers report the best (second best) performers.

## D.6 Effect of using cost-guided scoring function on baselines

In Tables 17 and 18, we report the impact of replacing the baselines' scoring function with our proposed cost-guided scoring function on three baselines across seven datasets for uniform and non-uniform cost settings, respectively. We notice that similar to the results reported in Section 5.2, the cost-guided scoring function helps the baselines perform significantly better in both the cost settings.

| | Mutag | Code2 | Molhiv | Molpcba | AIDS | Linux | Yeast |
|---|---|---|---|---|---|---|---|
| GMN-Match | 0.797 ± 0.013 | 1.677 ± 0.187 | 1.318 ± 0.020 | 1.073 ± 0.011 | 0.821 ± 0.010 | 0.687 ± 0.088 | 1.175 ± 0.013 |
| GMN-Match * | **0.654 ± 0.011** | **0.960 ± 0.092** | **1.008 ± 0.011** | **0.858 ± 0.009** | **0.601 ± 0.007** | **0.590 ± 0.084** | **0.849 ± 0.009** |
| GMN-Embed | 1.032 ± 0.016 | 1.358 ± 0.104 | 1.859 ± 0.020 | 1.951 ± 0.020 | 1.044 ± 0.013 | 0.736 ± 0.102 | 1.767 ± 0.021 |
| GMN-Embed * | **1.011 ± 0.017** | **1.179 ± 0.098** | **1.409 ± 0.015** | **1.881 ± 0.019** | **0.849 ± 0.010** | **0.577 ± 0.094** | **1.600 ± 0.017** |
| GREED | **1.398 ± 0.033** | 1.869 ± 0.140 | 1.708 ± 0.017 | **1.550 ± 0.017** | **1.004 ± 0.012** | 1.331 ± 0.169 | **1.423 ± 0.015** |
| GREED * | 2.133 ± 0.037 | **1.850 ± 0.156** | **1.644 ± 0.019** | 1.623 ± 0.017 | 1.143 ± 0.015 | **1.297 ± 0.151** | 1.440 ± 0.016 |
| GRAPHEDX | 0.492 ± 0.007 | 0.429 ± 0.036 | 0.781 ± 0.008 | 0.764 ± 0.007 | 0.565 ± 0.006 | 0.354 ± 0.043 | 0.717 ± 0.007 |

Table 17: Impact of cost-guided distance on MSE in uniform cost setting ($b^{\ominus} = b^{\oplus} = a^{\ominus} = a^{\oplus} = 1$). * represents the variant of the baseline with cost-guided distance. Green shows the best performing model. **Bold** font indicates the best variant of the baseline.

| | Mutag | Code2 | Molhiv | Molpcba | AIDS | Linux | Yeast |
|---|---|---|---|---|---|---|---|
| GMN-Match | 69.210 ± 0.883 | 13.472 ± 0.970 | 76.923 ± 0.862 | 23.985 ± 0.224 | 31.522 ± 0.513 | 21.519 ± 2.256 | 63.179 ± 1.127 |
| GMN-Match * | **1.592 ± 0.027** | **2.906 ± 0.285** | **2.162 ± 0.024** | **1.986 ± 0.021** | **1.434 ± 0.017** | **1.596 ± 0.211** | **2.036 ± 0.022** |
| GMN-Embed | 72.495 ± 0.915 | 13.425 ± 1.035 | 78.254 ± 0.865 | 28.437 ± 0.268 | 33.221 ± 0.523 | 20.591 ± 2.136 | 60.949 ± 0.663 |
| GMN-Embed * | **2.368 ± 0.039** | **3.272 ± 0.289** | **3.413 ± 0.037** | **4.286 ± 0.043** | **2.046 ± 0.025** | **1.495 ± 0.200** | **3.850 ± 0.042** |
| GREED | 68.732 ± 0.867 | 11.095 ± 0.773 | 78.300 ± 0.795 | 26.057 ± 0.238 | 34.354 ± 0.557 | 20.667 ± 2.140 | 60.652 ± 0.704 |
| GREED * | **2.456 ± 0.040** | **5.429 ± 0.517** | **3.827 ± 0.043** | **3.807 ± 0.040** | **2.282 ± 0.028** | **2.894 ± 0.394** | **3.506 ± 0.038** |
| GRAPHEDX | 1.134 ± 0.016 | 1.478 ± 0.118 | 1.804 ± 0.019 | 1.677 ± 0.016 | 1.252 ± 0.014 | 0.914 ± 0.110 | 1.603 ± 0.016 |

Table 18: Impact of cost-guided distance on MSE in non-uniform cost setting ($b^{\ominus} = 3, b^{\oplus} = 1, a^{\ominus} = 2, a^{\oplus} = 1$). * represents the variant of the baseline with cost-guided distance. Green shows the best performing model. **Bold** font indicates the best variant of the baseline.

## D.7 Results on performance of the alternate surrogates for GED

In Table 19, we present the performance of the alternate surrogates scoring function for GED discussed in B under non-uniform cost settings ($b^{\ominus} = 3, b^{\oplus} = 1, a^{\ominus} = 2, a^{\oplus} = 1$). From the results, we can infer that the alternate surrogates have comparable performance to GRAPHEDX however GRAPHEDX outperforms it by a small margin on six out of the seven datasets.

| | Mutag | Code2 | Molhiv | Molpcba | AIDS | Linux | Yeast |
|---|---|---|---|---|---|---|---|
| MAX-OR | 1.194 ± 0.016 | 1.112 ± 0.084 | 1.987 ± 0.022 | 1.806 ± 0.017 | 1.347 ± 0.014 | 1.009 ± 0.132 | 1.686 ± 0.018 |
| MAX | 1.351 ± 0.018 | 1.772 ± 0.122 | 1.972 ± 0.021 | 1.764 ± 0.017 | 1.346 ± 0.015 | 1.435 ± 0.169 | 1.748 ± 0.018 |
| GRAPHEDX | 1.134 ± 0.016 | 1.478 ± 0.118 | 1.804 ± 0.019 | 1.677 ± 0.016 | 1.252 ± 0.014 | 0.914 ± 0.110 | 1.603 ± 0.016 |

Table 19: Comparison of MSE between GRAPHEDX MAX-OR, and MAX. Green (yellow) numbers report the best (second best) performers.

## D.8 Comparison of zero-shot performance on other datasets

In Table 20, we compare all baselines with GRAPHEDX on zero-shot GED prediction on a new dataset. For each method, we select the best-performing models for {AIDS, Yeast, Mutag, Molhiv }, and test each one on the AIDS dataset under non-uniform cost setting.

| Train data | GRAPHEDX | Match | Embed | ISONET | GREED | ERIC | SimGNN | H2MN | GraphSim | EGSC |
|---|---|---|---|---|---|---|---|---|---|---|
| AIDS | 1.252 | 31.522 | 33.221 | 5.513 | 34.354 | 1.581 | 4.316 | 3.105 | 5.266 | 1.693 |
| Yeast | 1.746 | 35.24 | 38.542 | 7.631 | 40.838 | 2.774 | 4.851 | 3.805 | 8.404 | 2.061 |
| Mutag | 2.462 | 33.918 | 38.624 | 7.311 | 34.936 | 4.100 | 5.68 | 5.117 | 7.292 | 4.305 |
| Molhiv | 2.127 | 35.138 | 38.482 | 14.806 | 38.705 | 2.936 | 4.525 | 4.274 | 6.201 | 2.444 |

Table 20: Comparison of MSE between GRAPHEDX and baselines on zero-shot GED prediction on the AIDS test dataset under non-uniform cost setting. Green (yellow) numbers report the best (second best) performers.

## D.9 Importance of node-edge consistency

GRAPHEDX enforces consistency between node and edge alignments by design. However, one might choose to enforce node-edge consistency through alignment regularization between independently learnt soft node and edge alignment. However, as shown in Figure 21, we notice that such non-constrained learning might lead to under-prediction or incorrect alignments. We demonstrate the importance of constraining the node-pair alignment $S$ with the node alignment $P$ by showing the mapping of nodes and edges between two graphs. The required edit operations for subfigure a) with the constrained $S$ are two node additions $\{e, f\}$, one edge deletion $(d, a)$, and three edge additions $\{(a, f), (e, d), (e, f)\}$. Assuming that each edit costs one, the true GED is 6. However, in subplot b), $S$ is not constrained, and the edit operations with the lowest cost are two node additions $\{e, f\}$ and two edge additions $\{(a, f), (e, f)\}$. This erroneously results in a GED of 4.

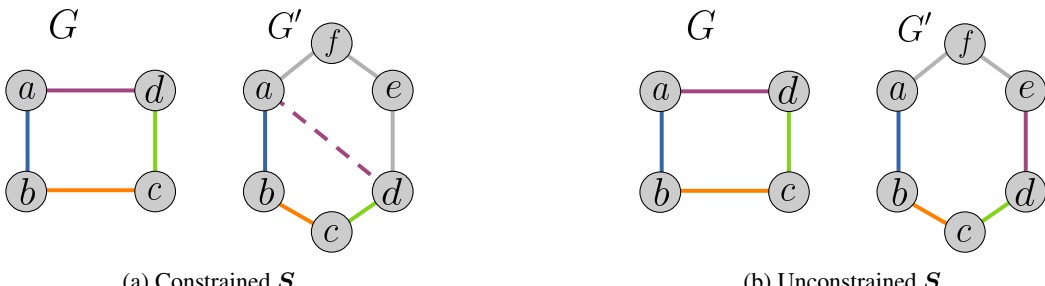

(a) Constrained $S$          (b) Unconstrained $S$

Figure 21: Node and edge alignment with constrained and unconstrained alignment $S$. A dashed edge represents the deleted edge. Grey edges represent added edges.

Further, in Table 22, we compare the performance of enforcing node-edge consistency through design (GRAPHEDX), and through alignment regularization (REG). Following the discussion in Section 4.2, such a model also exhibits a variant with XOR, called REG-xor. We notice that GRAPHEDX even outperforms such the described model in 4 out of 6 cases. We also notice that REG-xor outperforms GRAPHEDX in the other two cases. However, the above example shows a tendency to learn wrong alignments which in turn gives wrong optimal edit paths.

| | GED with uniform cost | | | GED with non-uniform cost | | |
|---|---|---|---|---|---|---|
| | Mutag | Code2 | Molhiv | Mutag | Code2 | Molhiv |
| REG | 0.536 | 0.576 | 0.848 | 1.162 | 1.488 | 1.877 |
| REG-xor | 0.513 | 0.587 | 0.826 | 1.309 | 1.440 | 1.711 |
| GRAPHEDX | 0.492 | 0.429 | 0.781 | 1.134 | 1.478 | 1.804 |

Table 22: Comparison of alignment regularizer usage versus no alignment regularizer usage on uniform cost GED, Measured by MSE. Green (yellow) numbers report the best (second best) performers.

### D.10 Comparison of nine possible combinations our proposed set distances

In Tables 23 and 24, we compare the performance of nine possible combinations our proposed set distances for uniform and non-uniform cost settings respectively. Results follow the observations in Table 2, where the variant with XOR-DIFFALIGN outperforms those without it.

| Edge edit | Node edit | Mutag | Code2 | Molhiv | Molpcba | AIDS | Linux | Yeast |
|---|---|---|---|---|---|---|---|---|
| DIFFALIGN | DIFFALIGN | 0.579 ± 0.0078 | 0.740 ± 0.0585 | 0.820 ± 0.0086 | 0.778 ± 0.0075 | 0.603 ± 0.0063 | 0.494 ± 0.0528 | 0.728 ± 0.0071 |
| DIFFALIGN | ALIGNDIFF | 0.557 ± 0.0073 | 0.742 ± 0.0612 | 0.806 ± 0.0088 | 0.779 ± 0.0076 | 0.597 ± 0.0063 | 0.452 ± 0.0614 | 0.747 ± 0.0078 |
| DIFFALIGN | XOR | 0.538 ± 0.0072 | 0.719 ± 0.0560 | 0.794 ± 0.0083 | 0.777 ± 0.0075 | 0.580 ± 0.0060 | 0.356 ± 0.0512 | 0.750 ± 0.0075 |
| ALIGNDIFF | DIFFALIGN | 0.537 ± 0.0072 | 0.513 ± 0.0367 | 0.815 ± 0.0085 | 0.773 ± 0.0074 | 0.606 ± 0.0064 | 0.508 ± 0.0607 | 0.731 ± 0.0073 |
| ALIGNDIFF | ALIGNDIFF | 0.578 ± 0.0079 | 0.929 ± 0.0659 | 0.833 ± 0.0086 | 0.773 ± 0.0075 | 0.593 ± 0.0062 | 0.605 ± 0.0678 | 0.761 ± 0.0076 |
| ALIGNDIFF | XOR | 0.533 ± 0.0074 | 0.826 ± 0.0565 | 0.812 ± 0.0083 | 0.780 ± 0.0074 | 0.575 ± 0.0060 | 0.507 ± 0.0568 | 0.889 ± 0.0138 |
| XOR | ALIGNDIFF | 0.492 ± 0.0066 | 0.429 ± 0.0355 | 0.788 ± 0.0084 | 0.766 ± 0.0074 | 0.565 ± 0.0062 | 0.416 ± 0.0494 | 0.730 ± 0.0072 |
| XOR | DIFFALIGN | 0.510 ± 0.0067 | 0.634 ± 0.0522 | 0.781 ± 0.0084 | 0.765 ± 0.0073 | 0.574 ± 0.0060 | 0.332 ± 0.0430 | 0.717 ± 0.0072 |
| XOR | XOR | 0.530 ± 0.0074 | 1.588 ± 0.1299 | 0.807 ± 0.0084 | 0.764 ± 0.0073 | 0.564 ± 0.0059 | 0.354 ± 0.0427 | 0.721 ± 0.0076 |
| GRAPHEDX | | 0.492 ± 0.0066 | 0.429 ± 0.0355 | 0.781 ± 0.0084 | 0.764 ± 0.0073 | 0.565 ± 0.0062 | 0.354 ± 0.0427 | 0.717 ± 0.0072 |

Table 23: Comparison of MSE for nine combinations of our neural set distance surrogates under uniform cost settings. The GRAPHEDX model was selected based on the best MSE on the validation set, while the reported results represent MSE on the test set. Green (yellow) numbers report the best (second best) performers.

| Edge edit | Node edit | Mutag | Code2 | Molhiv | Molpcba | AIDS | Linux | Yeast |
|---|---|---|---|---|---|---|---|---|
| DIFFALIGN | DIFFALIGN | 1.205 ± 0.0159 | 2.451 ± 0.2141 | 1.855 ± 0.0197 | 1.825 ± 0.0178 | 1.417 ± 0.0146 | 0.988 ± 0.1269 | 1.630 ± 0.0161 |
| DIFFALIGN | ALIGNDIFF | 1.211 ± 0.0164 | 2.116 ± 0.1581 | 1.887 ± 0.0199 | 1.811 ± 0.0174 | 1.319 ± 0.0140 | 1.078 ± 0.1168 | 1.791 ± 0.0185 |
| DIFFALIGN | XOR | 1.146 ± 0.0154 | 1.896 ± 0.1487 | 1.802 ± 0.0188 | 1.822 ± 0.0176 | 1.381 ± 0.0148 | 1.049 ± 0.1182 | 1.737 ± 0.0172 |
| ALIGNDIFF | DIFFALIGN | 1.185 ± 0.0159 | 1.689 ± 0.1210 | 1.874 ± 0.0202 | 1.758 ± 0.0169 | 1.391 ± 0.0145 | 0.914 ± 0.1099 | 1.643 ± 0.0163 |
| ALIGNDIFF | ALIGNDIFF | 1.338 ± 0.0178 | 1.488 ± 0.1222 | 1.903 ± 0.0204 | 1.859 ± 0.0179 | 1.326 ± 0.0141 | 1.258 ± 0.1335 | 1.731 ± 0.0171 |
| ALIGNDIFF | XOR | 1.196 ± 0.0164 | 1.741 ± 0.1151 | 1.870 ± 0.0196 | 1.815 ± 0.0174 | 1.374 ± 0.0146 | 1.128 ± 0.1330 | 1.802 ± 0.0194 |
| XOR | ALIGNDIFF | 1.134 ± 0.0158 | 1.478 ± 0.1178 | 1.872 ± 0.0202 | 1.742 ± 0.0168 | 1.252 ± 0.0136 | 1.073 ± 0.1211 | 1.639 ± 0.0162 |
| XOR | DIFFALIGN | 1.148 ± 0.0157 | 1.489 ± 0.1220 | 1.804 ± 0.0192 | 1.757 ± 0.0171 | 1.340 ± 0.0140 | 0.931 ± 0.1149 | 1.603 ± 0.0160 |
| XOR | XOR | 1.195 ± 0.0172 | 2.507 ± 0.1979 | 1.855 ± 0.0195 | 1.677 ± 0.0161 | 1.319 ± 0.0141 | 1.193 ± 0.1490 | 1.638 ± 0.0169 |
| GRAPHEDX | | 1.134 ± 0.0158 | 1.478 ± 0.1178 | 1.804 ± 0.0192 | 1.677 ± 0.0161 | 1.252 ± 0.0136 | 0.914 ± 0.1099 | 1.603 ± 0.0160 |

Table 24: Comparison of MSE for nine combinations under non-uniform cost settings. The GRAPHEDX model was selected based on the best MSE on the validation set, while the reported results represent MSE on the test set. Green (yellow) numbers report the best (second best) performers.

## D.11    Comparison of performance of our model with baselines using scatter plot

In Figure 25, we illustrate the performance of our model compared to the second-best performing model, under both uniform and non-uniform cost settings, by visualizing the distribution of outputs of the predicted GEDs by both models. We observe that predictions from our model consistently align closer to the $y = x$ line across various datasets showcasing lower output variance as compared to the next best-performing model.

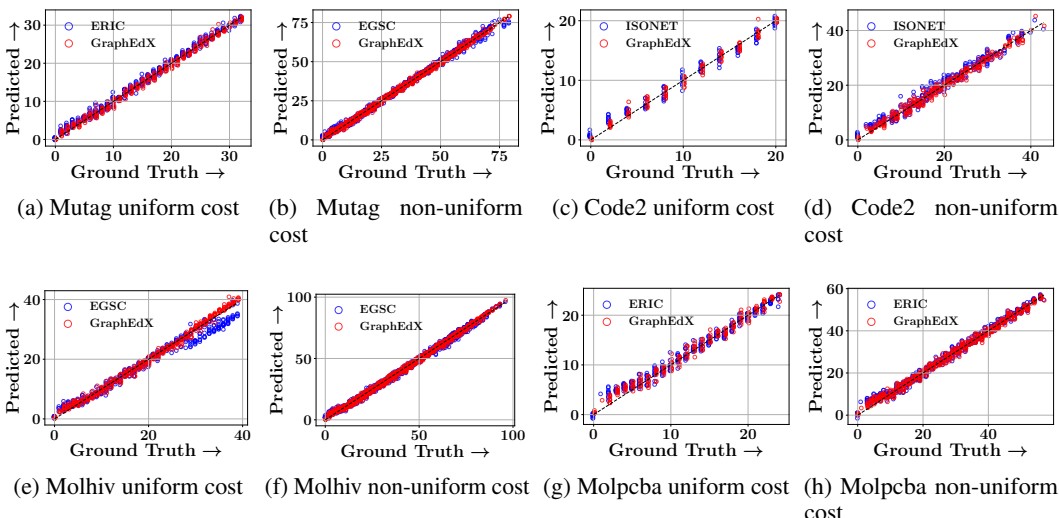

(a) Mutag uniform cost   (b) Mutag non-uniform cost   (c) Code2 uniform cost   (d) Code2 non-uniform cost

(e) Molhiv uniform cost   (f) Molhiv non-uniform cost   (g) Molpcba uniform cost   (h) Molpcba non-uniform cost

Figure 25: Scatter plot comparing the distribution of the predicted GED of our model with the next best-performing model across various datasets under both uniform and non-uniform cost settings.

## D.12    Comparison of performance of our model with baselines using error distribution

In Figure 26, we plot the distribution of error (MSE) of our model against the second-best performing model, under both uniform and non-uniform cost settings. We observe that our model performs better, exhibiting a higher probability density for lower MSE values and a lower probability density for higher MSE values.

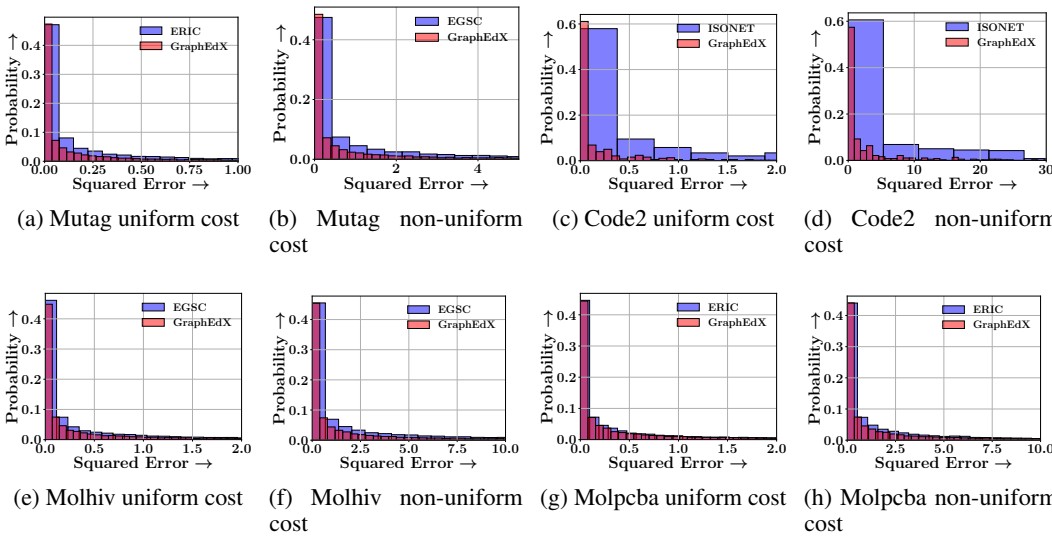

(a) Mutag uniform cost   (b) Mutag non-uniform cost   (c) Code2 uniform cost   (d) Code2 non-uniform cost

(e) Molhiv uniform cost   (f) Molhiv non-uniform cost   (g) Molpcba uniform cost   (h) Molpcba non-uniform cost

Figure 26: Error distribution of our model compared to the next best-performing model across various datasets under both uniform and non-uniform cost settings.

## D.13 Comparison of combinatorial optimisation gadgets for GED prediction

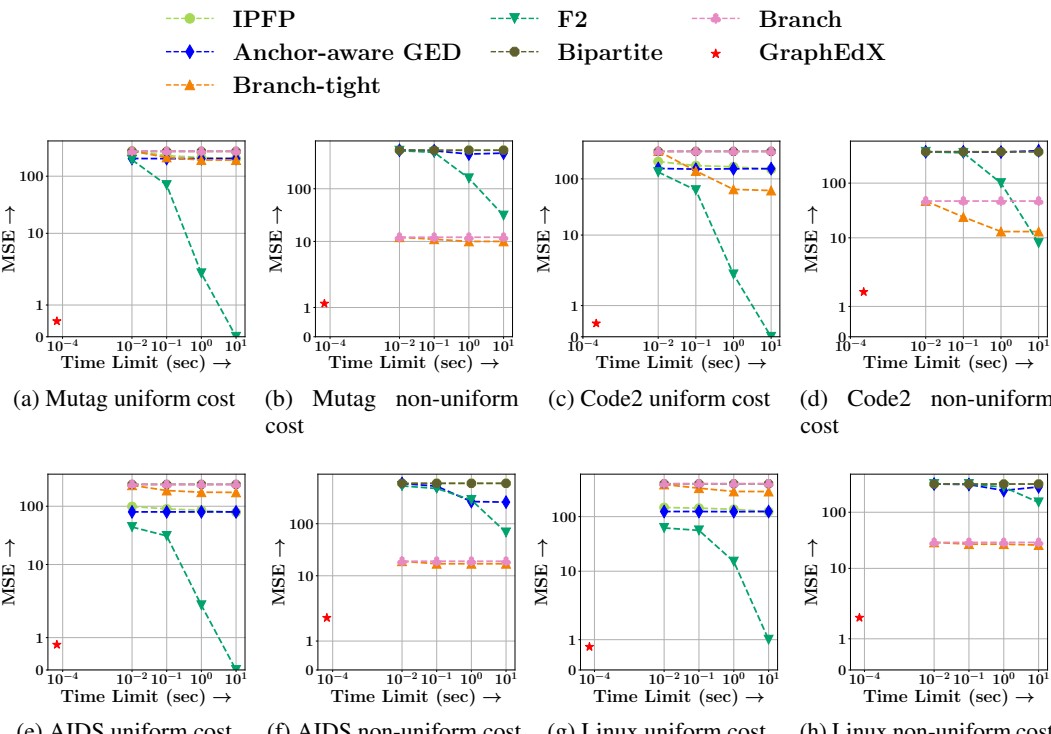

Figure 27: Performance of combinatorial optimization algorithms on various datasets under both uniform and non-uniform cost settings is evaluated. We plot MSE against the time limit allocated to the combinatorial algorithms. Additionally, we include the amortized time of our model and its MSE.

We compare the runtime performance of six combinatorial optimization algorithms described in Appendix C (ipfp [11], anchor-aware GED [15], branch tight [8], F2 [29], bipartite [41] and branch [8]). We note that combinatorial algorithms are slow to approximate the GED between two graphs. Specifically, GRAPHEDX often predicts the GED in $\sim 10^{-4}$ seconds per graph, however, the performance of the combinatorial baselines are extremely poor under such a time constraint. Hence, we execute the combinatorial algorithms with four different time limits per graph: ranging from $10^{-2}$ seconds (100x our method) to 10 seconds ($10^5$x our method).

In Figure 27, we depict the MSE versus time limit for the aforementioned combinatorial algorithms under both uniform and non-uniform cost settings. We also showcase the inference time per graph of our method in the figure. It is evident that even with a time limit scaled by $10^5$x, most combinatorial algorithms struggle to achieve a satisfactory approximation for the GED.

## D.14 Prediction timing analysis

In Figure 28 illustrates the inference time per graph of our model versus under uniform cost settings, averaged over ten runs. From the figure, we observe the following (1) GRAPHEDX outperforms four of the baselines in terms of inference time, and is comparable to ISONET's inference time (2) GMN-Embed, GREED, ERIC, and EGSC run faster compared to all other methods due to lack of interaction between graphs, which results in poorer performance at predicting the GED.

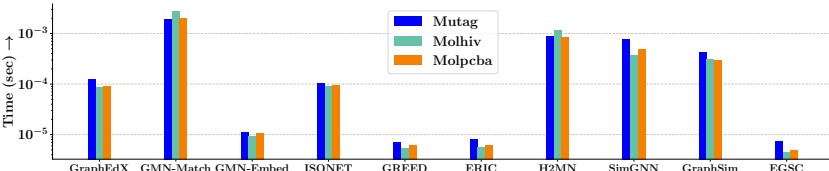

Figure 28: GED inference time comparison between our model and baselines. We notice that GRAPHEDX is consistently the third-fastest amongst all baselines. Although GMN-Embed and GREED have the lowest inference time, GRAPHEDX has much lower MSE consistently.

## D.15 Visualization (optimal edit path) + Pseudocode

In Algorithm 1, we present the pseudocode to generate the optimal edit path given the learnt node and edge alignments from GRAPHEDX. Figure 29 demonstrates how the operations in the edit path can be utilized to convert $G$ to $G'$.

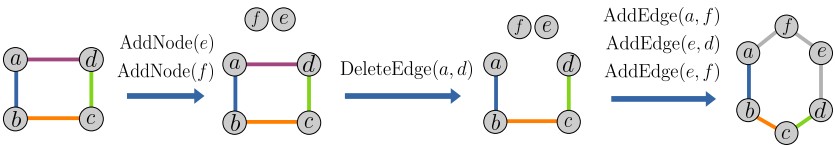

Figure 29: An example of the sequence of edit operations performed to convert one graph into another.

---

**Algorithm 1** Generation of Edit Path

---

1: **function** GETEDITPATH($G, G', \boldsymbol{\eta}_G, \boldsymbol{\eta}_{G'}$)
2:     $\boldsymbol{P}, \boldsymbol{S} \leftarrow$ GRAPHEDX($G, G', \boldsymbol{\eta}_G, \boldsymbol{\eta}_{G'}$)
3:     $\boldsymbol{P}, \boldsymbol{S} \leftarrow$ HUNGARIAN($\boldsymbol{P}$), HUNGARIAN($\boldsymbol{S}$)
4:     $o =$ NewList()
5:     **for** $(u, v) \in [N] \times [N]$ **do**
6:       **if** $\boldsymbol{P}[u, v] = 1$ and $\boldsymbol{\eta}_G[u] = 0$ and $\boldsymbol{\eta}_{G'}[v] = 1$ **then**
7:         AddItem($o$, ADDNODE($u$))
8:     **for** $(u, v), (u', v') \in \{[N] \times [N]\} \times \{[N] \times [N]\}$ **do**
9:       **if** $\boldsymbol{S}[(u, v), (u', v')] = 1$ and $\boldsymbol{A}[u, v] = 0$ and $\boldsymbol{A}'[u', v'] = 1$ **then**
10:         AddItem($o$, ADDEDGE($(u, v)$))
11:       **if** $\boldsymbol{S}[(u, v), (u', v')] = 1$ and $\boldsymbol{A}[u, v] = 1$ and $\boldsymbol{A}'[u', v'] = 0$ **then**
12:         AddItem($o$, DELEDGE($(u, v)$))
13:     **for** $(u, v) \in [N] \times [N]$ **do**
14:       **if** $\boldsymbol{P}[u, v] = 1$ and $\boldsymbol{\eta}_G[u] = 1$ and $\boldsymbol{\eta}_{G'}[v] = 0$ **then**
15:         AddItem($o$, DELNODE($u$))
16:     **return** $o$

---

