# OpenReview forum: "Graph Edit Distance with General Costs Using Neural Set Divergence"
_NeurIPS.cc/2024/Conference — NeurIPS 2024 poster_

### Official Review · Reviewer_81fW · 2024-06-24

**Soundness:** 3
**Presentation:** 2
**Contribution:** 3
**Rating:** 4
**Confidence:** 4

**Summary:**

This paper studies the approximation of non-uniform Graph Edit Distance (GED) using graph neural networks. It considers four types of edit operations with different costs: edge deletion, edge addition, node deletion, and node addition. The paper proposes explicitly accounting for the costs assigned to each type of edit operation to adapt to both uniform and non-uniform cost settings. Moreover, it introduces node-pair embeddings for connected and disconnected node pairs within graphs as edge and non-edge embeddings. It computes the node and node-pair alignments through three types of distance functions to approximate the considered operations. Experiments on real-world datasets under a variety of edit cost settings show that the proposed method outperforms selected baselines on chosen metrics.

**Strengths:**

* This paper studies an important task with many real-world applications.
* The authors propose generating learnable embeddings for disconnected node pairs, which is new to me.
* Extensive experiments have been conducted.

**Weaknesses:**

* The clarity of the paper should be improved. For example:
  * What are the differences between the pad indicator $q$ and  $η$? Are they inconsistent notation?
  * The paper is not self-contained. The related work should be in the main paper rather than in the appendix. Additionally, some related work, such as [1], which clearly highly influences this paper, is missing.
* The technical contribution of this paper is somewhat limited, as it directly uses MPNNs for representing nodes and the Gumbel-Sinkhorn network to establish alignment.
* The size of the graphs used in the experiments is relatively small (with at most 20 nodes) compared with the graphs used in other neural approaches.
* Restricting the model parameters for the baselines does not seem fair to me, since the proposed model heavily relies on the iterative non-parameterized Gumbel-Sinkhorn refinement.
* The size of the graphs used in the experiments is relatively small, so it is unclear if the proposed model can scale to graphs with, say, one or two hundred nodes. Moreover, since obtaining training data for GED is nontrivial, it is also unclear how the proposed model performs when the supervision signal is not the optimal GED value. I suggest that the authors conduct experiments on commonly used GED datasets with larger graphs, such as the IMDB dataset provided in [2], or use large synthetic graphs following [3].


Minors:

* There are some typos (Line 278 Additiional -> Additional). Some sentences are missing periods, for example, in Lines 273, 551.
* The caption of tables should be above the tables.

Reference

[1] Piao C, Xu T, Sun X, et al. Computing Graph Edit Distance via Neural Graph Matching[J]. Proceedings of the VLDB Endowment, 2023, 16(8): 1817-1829.

[2] Yunsheng Bai, Haoyang Ding, Song Bian, Ting Chen, Yizhou Sun, and Wei Wang. 2019. SimGNN: A Neural Network Approach to Fast Graph Similarity Computation. Proceedings of the Twelfth ACM International Conference on Web Search and Data Mining (2019).

[3] Roy, Indradyumna et al. “Maximum Common Subgraph Guided Graph Retrieval: Late and Early Interaction Networks.  NeurIPS 2022.

**Questions:**

* To what extent does the performance of the proposed model rely on the number of refinement iterations of the Gumbel-Sinkhorn network? Can the authors provide an analysis in this respect?
* To incorporate non-uniform edit costs into baselines, the author include the edit costs as initial features in the graphs. However, most baselines such as Eric [1] and SimGNN [2] only rely on the supervision signal during training. I wonder whether such an initialization is necessary and beneficial.
* The results regarding the inference time in Figure 26 seem illogical to me. Given that the complexity of Eric [1] is $\mathcal{O}(m + d_{ms}d^{′}T )$ in the inference stage (see the original paper), while the proposed method is at least $\mathcal{O}(k \cdot N^2)$, where $k$ is the number of refinement iterations, $N$ is the number of nodes. Can the authors provide the complexity analysis of the propose method?


References

[1] Wei Zhuo and Guang Tan. 2022. Efficient Graph Similarity Computation with Alignment Regularization. In Advances in Neural Information Processing Systems, Alice H. Oh, Alekh Agarwal, Danielle Belgrave, and Kyunghyun Cho (Eds.). https://openreview.net/forum?id=lblv6NGI7un

[2] Yunsheng Bai, Haoyang Ding, Song Bian, Ting Chen, Yizhou Sun, and Wei Wang. 2019. SimGNN: A Neural Network Approach to Fast Graph Similarity Computation. Proceedings of the Twelfth ACM International Conference on Web Search and Data Mining (2019).

---

> ### Author Rebuttal · Authors · 2024-08-07
>
> >  *differences between  𝑞 and 𝜂*
>
> Yes, we apologize for the mistake, $q$ = 1-$\eta$. The notation was redundant; only one is needed -- we will correct it.
> >  *some related work [1] is missing.*
>
> Since NeurIPS allowed including the Appendix with the main paper, we placed the related work there. If accepted, we will use the extra page to move the related work into the main paper.
> We also evaluate our method against GEDGNN [1] in both equal and unequal cost setups, below.
>
>
> |MSE||Mutag|Code2|Molhiv|Molpcba|Aids|Linux|Yeast|
> |:-|:-|:-|:-|:-|:-|:-|:-|:-|
> |Equal Cost|GEDGNN|33.281|31.124|39.163|12.419|20.006|10.521|32.128|
> |Equal Cost|GRAPHEDX|0.492|0.429|0.781|0.764|0.656|0.354|0.717|
> |Unequal Cost|GEDGNN|152.299|107.62|163.377|73.872|65.646|27.611|163.26|
> |Unequal Cost|GRAPHEDX|1.134|1.478|1.804|1.677|1.252|0.914|1.603|
>
> We observe that GraphEDX outperforms GEDGNN across all datasets. GEDGNN relies on node-level alignment supervision, which is unavailable in our setup, thereby degrading its performance. Additionally, as expected, GEDGNN, which is not designed to handle unequal cost setups, performs better at minimizing MSE in equal cost setups.
>
> >  *technical contribution.*
>
> GraphEdX is eventually simple and intuitive to describe, yet the judicious combination of MPNN and Sinkhorn networks significantly improves inductive bias. Our work is the first neural GED model that accounts for generic costs, requiring a deep understanding of GED costs and careful design of surrogate functions. Our experiments show that our method performs particularly well in non-uniform edit cost settings, highlighting the effectiveness of our set divergence guided scoring. Additionally, our method provides clear guidance on combining embeddings and alignments (DiffAlign, AlignDiff, XOR-DiffAlign) and uniquely enforces consistency between node and edge permutations, which is a significant contribution to the community.
>
> >  *The size of the graphs  is small.*
>
>  Please refer to global response.
> >  *#parameters for the baselines*
>
> In the table below, we compare the model performances on the Mutagenicity dataset under equal cost, across the original number of parameters and our reduced parameter setting. Performance is evaluated in terms of MSE, with the numbers in parentheses indicating the number of parameters. We observe that GraphEDX outperforms all baseline models across both parameter settings. Additionally, in most cases, the baselines perform comparably across both settings, except for GREED in the equal cost setting, where there is approximately a 40% reduction in error, albeit at the cost of a 44x increase in the number of parameters. Table (B) in the global PDF provides comparisons on other baselines, for both equal and unequal costs.
> |Mutagenicity|GraphEDX|GMN-Match|GREED|ERIC|H2MN|EGSC|
> |-|-|-|-|-|-|-|
> |Our|**0.492** (3030)|0.797 (2270)|1.398 (2434)|**0.719** (5902)|1.278 (2974)|0.765 (4248)|
> |Original|**0.492** (3030)|**0.686** (19936)|**0.807** (107776)|0.975 (45509)|**1.147** (18103)|**0.760** (54753)|
>
>
> Benchmarking across different baselines is challenging due to the large variation in the number of parameters (e.g., GREED has 2x more parameters than ERIC). Performance can vary with changes in the depth and width of different layers, making it difficult to create a standardized comparison platform without keeping the embedding dimensions and the number of GNN layers the same across all baselines. For example, in the restricted parameter setting, ERIC has a lower MSE than GREED, while in the original parameter setting, where GREED uses 2x the parameters of ERIC, it has a lower MSE than ERIC. This could be due to capturing spurious correlations rather than improving inductive bias.
>
>
> >  *analysis on Sinkhorn iterations*
>
> The following table shows the effect of varying the number of Sinkhorn iterations, under the equal cost setting on the Mutag dataset.
>
> |#Sinkhorn Iterations|1|2|3|4|8|12|16|20|
> |:-:|-|-|-|:-:|:-:|:-:|:-:|:-:|
> |MSE|13.531|7.889|2.42|1.422|0.684|0.529|0.498|0.492|
>
> We observe that within 12 iterations, it reaches 93% of the optimal MSE reported at 20 iterations.
>
> >  *edit costs as initial features*
>
> In order to ensure high-quality benchmarking, we allowed all baselines to leverage the knowledge of edit costs as much as possible, as they were not primarily designed for non-uniform edit costs. We had believed that this improves their inductive bias.
> As suggested by the reviewer, we performed an ablation study on the initialization of different baselines. The following table shows the results for Mutag and code2 (Fig D in global-pdf shows the rest). We observed that there is no consistent winner among different type of initializations. For ERIC and SIMGNN, in majority of cases, the setting without cost initialization is very marginally better than cost based initialization. GraphEdX outperforms all baselines.
> In the final draft, we will take the best performance across both settings for each baseline.
>
> ||Mutag|Code2|
> |-|-|-|
> |ERIC (init)|1.981|12.767|
> |ERIC (no-init)|**1.912**|**12.391**|
> |SimGNN (init)|4.747|**5.212**|
> |SimGNN (no-init)|**2.991**|8.923|
> |EGSC (init)|**1.758**|**3.957**|
> |EGSC (no-init)|1.769|4.395|
> |GraphEDX|1.134|1.478|
>
> >  *inference time in Figure 26 seem illogical*
>
> We are thankful to the reviewer for spotting this.  Upon closer inspection, we found that the inference times reported in Figure 26 included data batching time. Methods like GMN-Match, EMN-Embed, ISONET, and GraphEDX use numpy-CPU-based batching, while others use the PyTorch Geometric Data batching API, which is slower for our datasets. This was unfair to ERIC and possibly other baselines. To ensure a fair evaluation, we removed the data batching time and tracked only the forward pass times for each model. The updated histograms in Figure E of the global PDF show that ERIC is one of the fastest baselines. Additionally, GraphEDX is significantly faster than GMN-Match and H2MN, and comparable to ISONET.

---

### Official Review · Reviewer_htic · 2024-07-09

**Soundness:** 4
**Presentation:** 4
**Contribution:** 3
**Rating:** 7
**Confidence:** 4

**Summary:**

This paper proposes GRAPHEDX, an innovative neural model that deftly handles Graph Edit Distance (GED) calculations with customizable edit costs. By representing graphs as rich sets of node and edge embeddings and using a Gumbel-Sinkhorn permutation generator, GRAPHEDX captures the subtle nuances of graph structure that influence edit distances. Experiments across diverse datasets demonstrate GRAPHEDX's superiority, with the model consistently outperforming state-of-the-art baselines by significant margins, especially in scenarios with unequal edit costs.

**Strengths:**

1. The handling of GED metric with unequal cost is a timely and under-explored topic. In real world, many use cases involving GED need to consider such unequal cost scenario.
2. Rigorous evaluation on both the accuracy and efficiency (runnign time) is performed, showing that the proposed method achieves high accuracy while maintaining efficiency.
3. The current model may not fully address richly attributed graphs with complex node and edge features.
To show the transfer learning ability, I encourage the authors to try training on one dataset and testing on another. This would reduce concern on overfitting to one particular dataset
The paper is well-written.

**Weaknesses:**

1. The current model may not fully address richly attributed graphs with complex node and edge features.
2. To show the transfer learning ability, I encourage the authors to try training on one dataset and testing on another. This would reduce concern on overfitting to one particular dataset

**Questions:**

N/A

---

> ### Author Rebuttal · Authors · 2024-08-07
>
> >  *The current model may not fully address richly attributed graphs with complex node and edge features.*
>
> We did not perform any experiments on node and edge features, but we can encode node feature and edge features in our models. Specifically, node features can be encoded during feature initialization and edge features can be encoded in the message passing step Eq. 24, page 22 as $LRL _{\theta} (x _k(u), x _k(v), \text{EdgeFeature} _{uv})$ instead of $LRL _{\theta} (x _k(u), x _k(v))$.
>
>
> >  *To show the transfer learning ability, I encourage the authors to try training on one dataset and testing on another. This would reduce concern on overfitting to one particular dataset.*
>
> In the following, we show the inference MSE of different models trained on the AIDS, Yeast, Mutag, and Molhiv datasets, when tested on the AIDS dataset. Analyzing the table column-wise, we observe the expected drop in performance across all methods when using models trained on datasets other than AIDS. Additionally, within each individual row, GraphEDX consistently outperforms all baselines, regardless of the training dataset. In Table C of the global pdf, we present comparison with all baselines.
> | Test  Dataset | Train Dataset | GraphEDX | ERIC  | SimGNN | H2MN  | EGSC  |
> |---------------|---------------|----------|-------|--------|-------|-------|
> | AIDS          | AIDS          | 1.252    | 1.581 | 4.316  | 3.105 | 1.693 |
> | AIDS          | Yeast         | 1.746    | 2.774 | 4.581  | 3.805 | 2.061 |
> | AIDS          | Mutag         | 2.462    | 4.1   | 5.68   | 5.117 | 4.305 |
> | AIDS          | Molhiv        | 2.127    | 2.936 | 4.525  | 4.274 | 2.444 |

---

> > ### Comment · Reviewer_htic · 2024-08-12
> >
> > Thank you for your response. I have read the response.

---

> > > ### Comment · Reviewer_eEwY · 2024-08-14
> > >
> > > Thank you for the detailed response.

---

### Official Review · Reviewer_eEwY · 2024-07-27

**Soundness:** 3
**Presentation:** 3
**Contribution:** 4
**Rating:** 7
**Confidence:** 4

**Summary:**

This paper proposes GRAPHEDX, a neural model that learns to estimate the graph edit distace among a pair of graphs, not only in the case of equal and symmetric costs for edit operations, but in the case of unequal costs specified for the four edit operations. The core of the proposal represents each graph as a set of node & edge embeddings, designs neural set divergence surrogates based on those embeddings, and replaces the QAP term corresponding to each operation with its surrogate. The method learns alignments to compute surrogates via a Gumbel-Sinkhorn permutation generator while also ensuring consistency between the node and edge alignments and rendering them sensitive to the presence and absence of edges between node-pairs.

**Strengths:**

S1. Addresses a gap in the field that previous works have left open.
S2. Outpeforms previous effort in both equal-cost and unequal-cost settings.

**Weaknesses:**

W1. The cost of training and estimating is not shown.

**Questions:**

Q: What is the cost of training and estimation?

**Limitations:**

Yes.

---

> ### Author Rebuttal · Authors · 2024-08-07
>
> > *What is the cost of training and estimation?*
>
> In Appendices F.11 and F.12, we discuss the computational cost of GED estimation. The following table presents the total training time and the amortized per-graph pair inference time (in milliseconds) for different methods on all dataset. The training time includes variations due to different numbers of training epochs for each method, under the same early stopping criteria. The inference time is averaged across all graph pairs and does not include data preprocessing times, which may vary across methods. In the global PDF, we provide the histogram of per-graph pair inference time (Figure E).
>
> ||Dataset| GRAPHEDX | GMN-Match | GMN-Embed | ISONET | GREED | ERIC  | SimGNN | H2MN  | GraphSim | EGSC  |
> |------|----|----------|-----------|-----------|--------|-------|-------|--------|-------|----------|-------|
> | Training Time (min) | Mutag | 422      | 465       | 44        | 238    | 50    | 1185  | 556    | 2268  | 415      | 166   |
> |                     | Code2   | 38       | 41        | 5         | 8      | 6     | 37    | 102    | 60    | 28       | 25    |
> |                     | Molhiv  | 479      | 877       | 58        | 379    | 95    | 1480  | 1058   | 1546  | 780      | 401   |
> |                     | Molpcba | 847      | 986       | 97        | 489    | 112   | 1605  | 1358   | 1125  | 1135     | 325   |
> |                     | AIDS         | 388      | 748       | 70        | 107    | 63    | 1179  | 1343   | 3123  | 705      | 426   |
> |                     | Linux        | 39       | 11        | 2         | 4      | 2     | 26    | 10     | 24    | 7        | 4     |
> |                     | Yeast        | 390      | 936       | 90        | 228    | 115   | 2801  | 642    | 1449  | 464      | 827   |
> | Inference Time (ms) | Mutag | 0.127    | 1.945     | 0.011     | 0.105  | 0.007 | 0.008 | 0.869  | 0.763 | 0.423    | 0.008 |
> |                     | Code2   | 0.257    | 1.638     | 0.16      | 0.28   | 0.169 | 0.205 | 0.862  | 0.449 | 0.465    | 0.189 |
> |                     | Molhiv  | 0.088    | 2.78      | 0.009     | 0.092  | 0.005 | 0.006 | 1.18   | 0.376 | 0.304    | 0.005 |
> |                     | Molpcba | 0.089    | 1.975     | 0.011     | 0.094  | 0.006 | 0.006 | 0.831  | 0.488 | 0.297    | 0.005 |
> |                     | AIDS         | 0.071    | 1.512     | 0.009     | 0.072  | 0.007 | 0.007 | 0.672  | 0.289 | 0.263    | 0.006 |
> |                     | Linux        | 0.51     | 1.782     | 0.357     | 0.471  | 0.349 | 0.477 | 1.275  | 0.627 | 0.705    | 0.374 |
> |                     | Yeast        | 0.073    | 1.537     | 0.012     | 0.076  | 0.008 | 0.008 | 0.65   | 0.297 | 0.266    | 0.007 |
>
> We observe that GraphEDX is generally faster in training than GMN-Match, ERIC, and H2MN, and is comparable to SimGNN and GraphSIM. In terms of inference, GraphEDX is significantly faster than GMN-Match with cost-guided distance, which outperforms all other baselines in terms of accuracy (see Table 3 vs. Table 2 in the main paper), but is slower than ERIC and EGSC. However, the efficiency of ERIC and EGSC comes at the cost of lower accuracy.

---

### Author Rebuttal · Authors · 2024-08-07

> *Evaluation on larger graphs*

As has been correctly pointed out, obtaining training data for GED is nontrivial for larger graphs, and baselines have often used noisy GED values for supervision. Since our work addresses asymmetric cost-sensitive GED for the first time, we preferred to use exact optimal GED values to better evaluate our proposal and the baselines. However, we agree that our proposal should also be evaluated with larger graphs.

We generate large graphs by randomly selecting graphs of size 60-80 from the AIDS dataset and appending them to graph pairs in our dataset. This results in graphs averaging 80 nodes, with a maximum of 100 nodes, introducing some noise to the unequal cost GED values.

The following table shows the results for our methods and the baselines, where we use the number of parameters originally specified by each baseline. We observe that GraphEDX has the lowest MSE, followed by ISONET, H2MN and ERIC. Table A in the global pdf reports numbers on all methods in this setting.
|                 | GraphEDX | GMN-Match | ISONET | ERIC   | H2MN   | EGSC   |
|-----------------|----------|-----------|--------|--------|--------|--------|
| MSE:  | 3.986    | 52.156    | 8.324  | 24.584 | 6.695 | 43.467 |


Moreover, as suggested by the reviewer, we evaluate the scalability on large graphs with more than 20 nodes. The following table presents the amortized inference time (in milliseconds) per graph pair, demonstrating that our method can easily scale to large graphs.
|Inference Time (ms)| \|V\|=20 | \|V\|=30 | \|V\|=50 | \|V\|=70 | \|V\|=100 | \|V\|=150 | \|V\|=200 |
|:-|:-|:-|:-|:--|:-|:-|:-|
|GRAPHEDX|0.165|0.165|0.200|0.290|0.768|4.616|10.485|


To further evaluate performance on larger graphs, we used a pretrained model on Ogbg-Code2 dataset graphs with fewer than $|V| = 20$ nodes and tested it on graphs with nodes  $|V| \in (20, 50]$  from the same dataset, with an average size of 30. The following table shows the results in terms of MSE under the unequal cost setup:
| | GraphEDX | GMN-Match | ISONET | SimGNN | H2MN  |
|-|-|-|-|-|--|
| Code2-(20,50] | 6.302 | 53.297 | 7.052  | 48.072 | 9.486 |

We observe that our method outperforms all the baselines, with ISONET and H2MN being the second and third best performers respectively. Table A in the gobal pdf reports numbers on the remaining baselines.

---

### Author Response · Authors · 2024-08-13
**Summary of Rebuttal and Thanks to Reviewers**

We are grateful for the detailed feedback from the reviewers. In response, we have provided the following clarifications:

**Cost of Training and Estimation:** We provided detailed information on the computational costs of GED estimation and presented the training and inference times for different datasets and methods. In particular, we appreciate Reviewer 81fW’s feedback on inference time measurements. Based on their input, we revised our estimates to exclude data batching time, resulting in more accurate comparisons. The updated results show that GraphEDX is significantly faster than GMN-Match and H2MN, and competitive with ISONET, providing a clearer picture of our method's efficiency.

**Scaling to Larger Graph Sizes:** We demonstrated scalability by training under noisy ground truth for larger graphs. Our method consistently outperformed the baselines, indicating robustness in handling larger, noisier datasets with more complex graphs.

**Comparison with Additional Baselines:** We compared GraphEDX with GEDGNN under both equal and unequal cost settings. GraphEDX significantly outperformed GEDGNN across all datasets, particularly in the unequal cost setups where GEDGNN is not specifically designed to excel.

**Feature Initialization and Number of Parameters in Baselines:** We addressed concerns regarding the number of parameters and initial features for the baselines. We compared model performances with both original and our parameter settings and conducted an ablation study on different initialization strategies. Our results show that GraphEDX consistently outperforms all baselines across these settings. We will update the numbers in the final draft to ensure that all baselines are given the best possible advantage under these configurations.

**Sensitivity to Sinkhorn Iterations:** We investigated the sensitivity of our method to the number of Sinkhorn iterations. Our results show that within 12 iterations, the method reaches 93% of the best performance earlier reported at 20 iterations, demonstrating that our approach converges efficiently, with a reliably diminishing returns profile.

We appreciate the reviewers' feedback, which can be readily incorporated in the final version of the paper.

---

### Decision · Program_Chairs · 2024-09-25

**Decision:**

Accept (poster)

**Comment:**

The paper proposes a neural-based graph edit distance (GED) estimator that can work with general costs specified for the four edit operations (edge deletion, edge addition, node deletion, and node addition). The method is evaluated on different combinations of datasets and edit cost settings, showing consistent improvement with respect to existing GED estimation heuristics.

This paper provides an ingenuous method to a NP-hard problem, which seems to work in practical applications. Furthermore, the authors positively reply to each (valid) concern raised by the reviewers during the rebuttal. I recommend acceptance.